# Role of laterodorsal tegmentum projections to nucleus accumbens in reward-related behaviors

Bárbara Coimbra [1,2], Carina Soares-Cunha[1,2], Nivaldo AP Vasconcelos[1,2,3], Ana Verónica Domingues[1,2], Sónia Borges[1,2], Nuno Sousa [1,2,4] & Ana João Rodrigues [1,2,4]

The laterodorsal tegmentum (LDT) is associated with reward considering that it modulates VTA neuronal activity, but recent anatomical evidence shows that the LDT also directly projects to nucleus accumbens (NAc). We show that the majority of LDT-NAc inputs are cholinergic, but there is also GABAergic and glutamatergic innervation; activation of LDT induces a predominantly excitatory response in the NAc. Non-selective optogenetic activation of LDT-NAc projections in rats enhances motivational drive and shifts preference to an otherwise equal reward; whereas inhibition of these projections induces the opposite. Activation of these projections also induces robust place preference. In mice, specific activation of LDT-NAc cholinergic inputs (but not glutamatergic or GABAergic) is sufficient to shift preference, increase motivation, and drive positive reinforcement in different behavioral paradigms. These results provide evidence that LDT-NAc projections play an important role in motivated behaviors and positive reinforcement, and that distinct neuronal populations differentially contribute for these behaviors.

[1] Life and Health Sciences Research Institute (ICVS), School of Medicine, University of Minho, 4710-057 Braga, Portugal. [2] ICVS/3B's–PT Government Associate Laboratory, Braga/Guimarães, Portugal. [3] Department of Biomedical Engineering, Federal University of Pernambuco, Recife, Pernambuco 50670-901, Brazil. [4] Clinical Academic Center (2CA-Braga), Braga, Portugal. Correspondence and requests for materials should be addressed to N.S. (email: njcsousa@med.uminho.pt) or to A.J.R. (email: ajrodrigues@med.uminho.pt)

Pharmacological, lesion, and optogenetic studies have involved the laterodorsal tegmentum (LDT) in reward processing and reinforcement[1–7]. The LDT sends specific inputs to the ventral tegmental area (VTA), that mainly target the dopaminergic population of this region[8,9]. LDT regulates firing activity of midbrain dopamine neurons and, consequently, nucleus accumbens (NAc) dopamine levels. For example, LDT electrical stimulation increases dopamine cell burst firing and boosts dopamine release in the NAc, whereas LDT lesions reduce dopamine burst firing[10–13].

Recent behavioral studies have shown that optogenetic stimulation of LDT-VTA neurons enhance place preference[3,14] and induce intracranial self-stimulation in rats[6,15]. However, LDT effects in reward-related behaviors may go beyond this direct control of VTA, since a neuroanatomical study has shown that the LDT can also directly innervate NAc[16]. LDT neurons topographically innervate wide areas of the striatum, preferentially projecting to the medial striatum and NAc core, forming mainly putative excitatory synapses. Around 60% of these projections are thought to be cholinergic, with an additional contribution of GABAergic and glutamatergic projections. Remarkably, to date, the functional role of LDT-NAc direct projections and how they contribute for behavior, and in particular for reward and reinforcement, has not been studied.

Here we show that the LDT mainly drives an excitatory cholinergic input to the NAc, but there is also glutamatergic and GABAergic innervation. We observed that non-selective optogenetic activation of LDT-NAc projections was sufficient to increase motivation, induce preference and positive reinforcement in rats, whereas inhibition produced the opposite outcome. We further showed that most of these effects were mediated by LDT-NAc cholinergic inputs, whereas optical modulation of glutamatergic and GABAergic inputs originated different behavioral outcomes.

## Results

**Anatomical and functional validation of LDT-NAc inputs.** To confirm previous findings showing that LDT sends direct projections to NAc[16], we injected in the NAc of rats an adeno-associated virus (AAV5) containing a vector encoding for wheat germ agglutinin cre fusion protein (AAV5–EF1a–WGA–Cre–mCherry) in combination with injection in the LDT of cre-dependent channelrhodopsin (AAV5-Ef1a-DIO-hChR2(H134R)-eYFP) (Fig. 1a). This allows ChR2 expression only in regions directly connected to the NAc. We found abundant YFP expression in the LDT, indicative of direct LDT-NAc projections. YFP expression was distributed throughout LDT neurons soma, dendrites and axons (Fig. 1b). YPF was also observed in LDT terminals in the NAc (Fig. 1c). We performed immunofluorescence (IF) using classical markers for cholinergic (choline acetyltransferase—ChAT), glutamatergic (excitatory amino-acid (glutamate) transporter—EAAC1) and GABAergic (glutamate decarboxylase 65+67—GAD) neurons. Nearly 50% of YFP-transfected LDT neurons were cholinergic, 29% glutamatergic, and 23% GABAergic (Fig. 1d, e); in line with previous observations showing that 59–74% of LDT-NAc projections were cholinergic[16].

In order to ensure minimal second order transynaptic migration of WGA-cre[16,17], we performed single-cell in vivo electrophysiological recordings 4 weeks post-injection (Fig. 1f). Optical activation (20 Hz, 80 10 ms light pulses) of LDT cells increased the net firing rate of this region (Fig. 1g; RM 1way ANOVA, $F_{(1.182, 28.36)} = 11.66$, $p = 0.0012$, $n = 25$ cells). Stimulation induced an increase in the firing rate of 48% of recorded neurons and 12% presented decreased activity (Fig. 1h).

To avoid indirect activation of other brain regions due to the existence of LDT collaterals[16], we also stimulated LDT terminals in the NAc. LDT terminal stimulation elicited a net increase in NAc firing rate (Fig. 1i; RM 1way ANOVA, $F_{(1.421, 90.96)} = 176.6$, $p < 0.0001$, $n = 65$ cells), with a short latency upon stimulation (Supplementary Fig. 4b). This evoked an excitatory response in 54% of cells, and 12% decreased activity during stimulation (Fig. 1j). The heatmap of percentage of firing rates (0.5s bins) of all recorded cells showed that a large portion of cells increased firing rate during optical stimulation (considering as a response more than 20% of change from baseline) (Fig. 1k). After stimulation, firing rates returned to baseline activity. Firing rate significantly differed from a 10-s baseline window to the 4-s stimulus (Fig. 1l; Kolmogorov-Smirnov test two tailed, $p < 0.001$). Based on neuronal waveform and firing rate characteristics[18,19], we further divided NAc cells into putative medium spiny neurons (pMSNs), cholinergic interneurons (pCIN), and fast-spiking GABAergic interneurons (pFS). We found that 52% of recorded pMSNs increased their firing rate upon LDT terminal activation (Fig. 1m). The majority of pCINS and pFS interneurons also showed an increase in firing rate upon LDT axon terminal stimulation, although these results should be interpreted with caution since we recorded a small number of these cells (Fig. 1m). Correct electrode placement was confirmed for all animals included in the electrophysiological analysis (Supplementary Fig. 1).

**LDT-NAc optical activation shifts preference and increases motivation.** Our next step was to evaluate whether LDT-NAc projections modulated reward-related behaviors (timeline in Supplementary Fig. 1). We used the same strategy as above for ChR2 animals (WGA–Cre in NAc + DIO-ChR2 in LDT), and included a control YFP group (WGA–Cre in NAc + DIO-YFP in LDT). In another set we also tested animals injected only with cre-dependent ChR2 in the LDT to ensure that ChR2 was not expressed in the absence of WGA-cre. Only animals with correct cannula placement were used (Supplementary Fig. 1).

Animals were tested in a two-choice lever operant task, in which pressing either lever gives a pellet reward, and one lever is arbitrarily selected to deliver the pellet with simultaneous LDT-NAc terminals' optogenetic stimulation (stim+; excitation: 473 nm laser, 80 10 ms pulses at 20 Hz) (Fig. 2a). We observe a significant effect of session and group (Fig. 2b; RM 2way ANOVA, session: $F_{(7, 476)} = 66.59$, $p < 0.0001$; group: $F_{(3, 68)} = 32.31$, $p < 0.0001$). Specifically, in the last session, no effect of stimulation was found in YFP animals, since they presented a similar number of presses in both levers. In contrast, LDT-NAc stimulation induced a clear preference for the stim+ lever in ChR2 animals (Fig. 2b; RM 2way ANOVA, stim+ vs stim-: $t_{(544)} = 15.28$, $p < 0.0001$), and a substantial increase in the number of lever presses in stimulated ChR2 animals when compared to YFP controls, with no alteration of the total number of lever presses (Supplementary Fig. 5a), suggesting increased motivation.

To evaluate if LDT-NAc stimulation was reinforcing per se or only when paired with an external reward, animals were tested in the same two-choice task, but in pellet extinction conditions (Fig. 2c). Pressing stim+ lever still yielded laser stimulation but no pellet is given; pressing the other lever also does not yield any pellet. Both groups decreased instrumental responding as early as the first session for both levers (RM 2way ANOVA, session: $F_{(4, 272)} = 219.2$, $p < 0.0001$). This indicates that the pairing of the stimulation with the reward is crucial for the positive reinforcing properties of LDT-NAc stimulation in operant behavior. Moreover, it also hints that the behavioral flexibility was not compromised, which was further confirmed in a reversal session in which the levers were switched (Supplementary Fig. 5c, d).

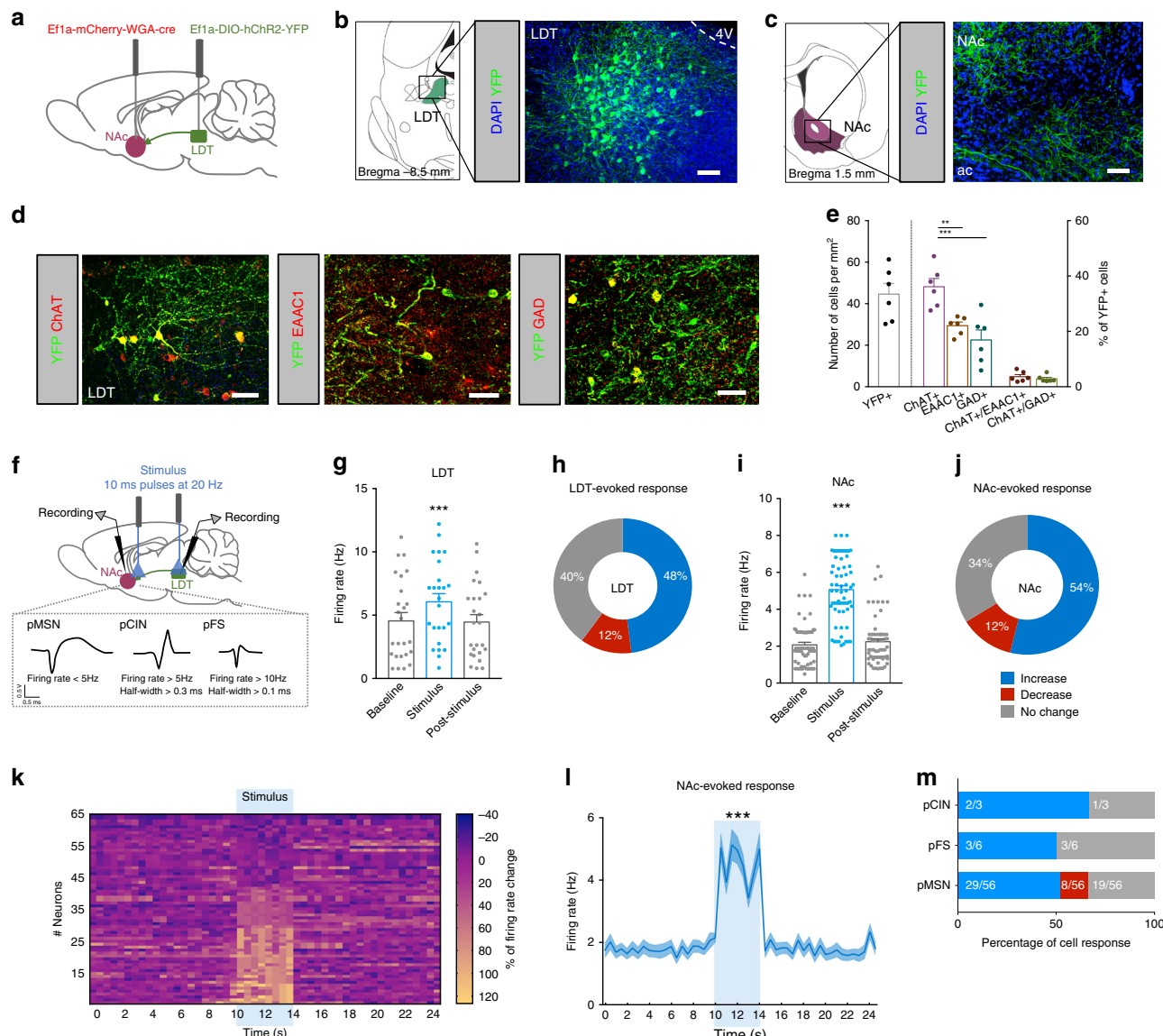

**Fig. 1** LDT stimulation drives a predominantly excitatory response in the NAc. **a** Strategy used for LDT-NAc optogenetic manipulation. **b** Representative immunofluorescence showing YFP staining in the LDT and **c** in terminals in the NAc; scale bar = 100 μm. **d** Representative immunofluorescence for eYFP (green) and ChAT, EAAC1 or GAD65/67 (red); scale bar = 50 μm. **e** Respective quantification of double and triple positive cells, indicative of transfected neurons ($n = 6$ animals, 1way ANOVA). **f** Electrophysiological strategy for cell recording in the LDT and in the NAc. NAc neurons were separated into pMSNs, pCINs, and pFS. **g** LDT neurons increase firing rate in response to optical activation of LDT cell bodies (80 pulses of 10ms at 20 Hz) ($n = 5$ animals; 25 LDT cells; 1way ANOVA). **h** 48% of LDT recorded cells increase their firing rate during stimulation. **i** NAc cells increase firing rate in response to LDT optical stimulation ($n = 9$ animals; 65 cells; 1way ANOVA). **j** Around half of recorded cells in the NAc show an increase in the firing rate upon stimulation, 34% present no change and 12% decrease activity. **k** Heatmap representation of percentage of cell responses in the NAc when LDT terminals are stimulated. Each row represents a neuron. **l** Average distribution of the firing rate of NAc neurons showing an increase in activity during stimulation period (KS test). **m** 52% of recorded pMSNs increase their activity (29/56 cells), 34% did not change firing rate (19/56 cells) and 14% decrease their activity (8/56 cells); 67% pCINs (2/3 cells) and 50% pFS (3/6 cells) interneurons increase and 33% pCINs (1/3 cells) and 50% pFS (3/6 cells) interneurons do not change their activity. Values are shown as mean ± s.e.m. **$p < 0.01$, ***$p < 0.001$. ac- anterior commissure; 4V: 4th ventricle

To evaluate persistence of preference for the LDT-NAc stimulation-associated lever, we performed the same task but in laser extinction conditions, which makes the outcome (one pellet) equal in both levers. ChR2 animals still manifest preference for the previously laser-associated lever (stim+), in spite of the lack of stimulation (Fig. 2d; RM 2way ANOVA, group: $F_{(3, 68)} = 27.4$, $p < 0.0001$).

Next, we performed the progressive ratio schedule of reinforcement that measures the willingness to work to get a food pellet. The breakpoint is a direct measure of motivation, and

is the maximum effort an animal reaches before giving up when the price of a reward increases substantially throughout a session (Fig. 2e). Session and group had a significant effect in the number of cumulative presses (Fig. 2f; RM 2way ANOVA, session: $F_{(5, 340)} = 330.7$, $p < 0.0001$; group: $F_{(3, 68)} = 19.12$, $p < 0.0001$). ChR2 animals presented increased cumulative presses in the stim+ lever (RM 2way ANOVA, stim+ vs stim-: post hoc $t_{(408)} = 11.46$, $p < 0.0001$); which was translated into a higher breakpoint for stim+ lever in the ChR2 group (Fig. 2g; RM 2way ANOVA, stim + vs stim-: post hoc $t_{(34)} = 15.2$, $p < 0.0001$), indicative of

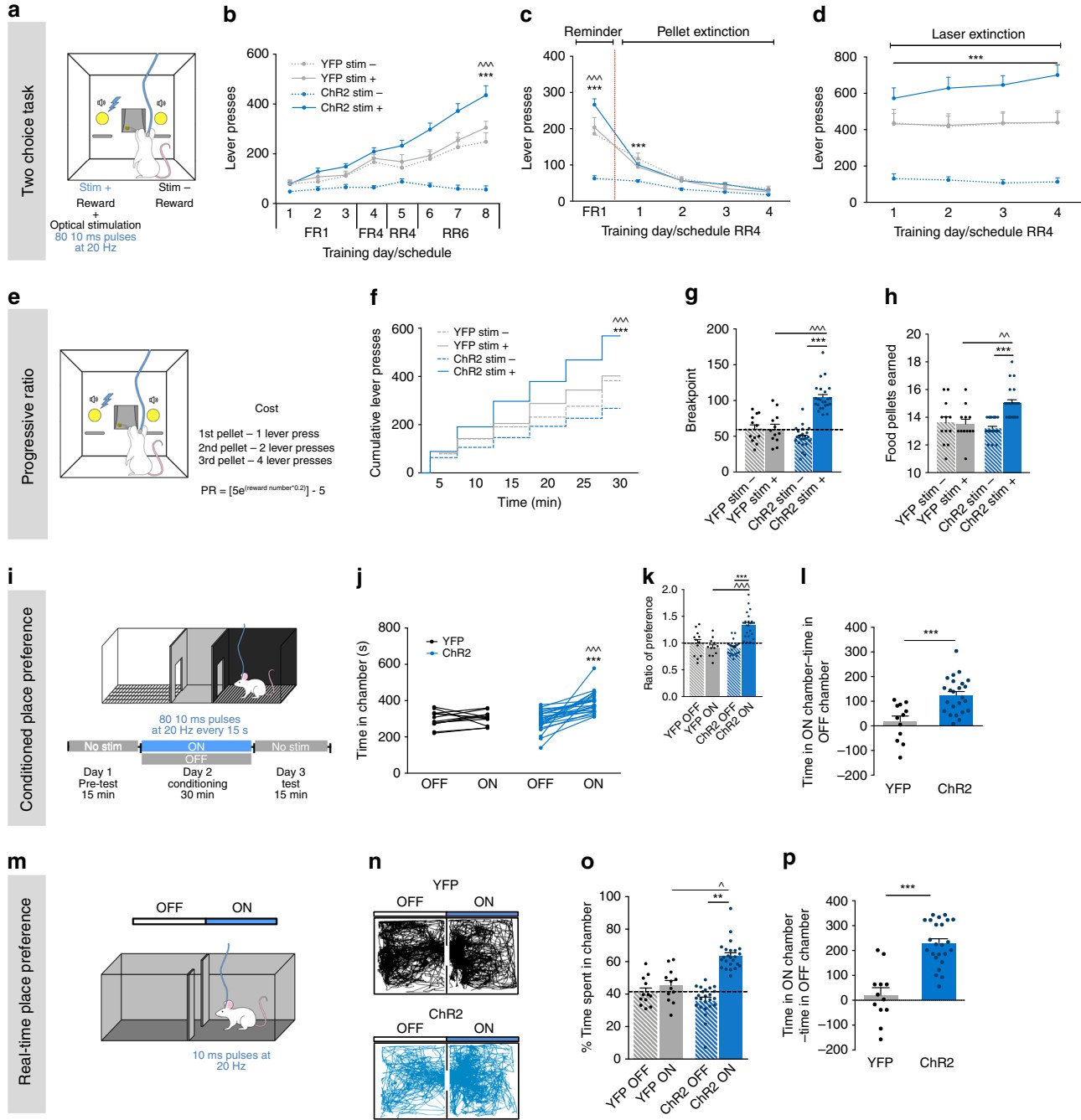

**Fig. 2** LDT-NAc optical activation induces preference and increases motivation. **a** Schematic representation of the two-choice task. Pressing stim- lever yields one food pellet and pressing stim+ lever delivers one pellet + optical stimulation of LDT-NAc inputs (80 10 ms pulses at 20 Hz). **b** Time-course representation of the responses in ChR2 ($n = 24$) and YFP ($n = 12$) rats. Optogenetic activation of LDT-NAc terminals focuses responses for the lever associated with the laser-paired reward (stim+) over an otherwise equivalent food reward (stim-) in ChR2 animals, but not in control YFP group. **c** In pellet extinction conditions, both groups decrease responses for both levers. **d** In laser extinction conditions, ChR2 animals still manifest preference for the stim+ lever, despite no stimulation being given; YFP animals do not manifest preference. **e** In the progressive ratio task animals have to increasingly press a lever more times to obtain the same reward (stimulation associated with stim+ lever). **f** Cumulative presses performed during the progressive ratio task show that ChR2 animals press more on stim+ lever. **g** Increase in breakpoint for stim+ lever in ChR2 animals, indicative of enhanced motivation. **h** Food pellets earned during the PR task. **i** CPP and **m** RTPP paradigms, in which one chamber is associated with laser stimulation (ON side). **j** Total time spent in the OFF and ON sides in YFP and ChR2 groups. **k** Ratio of preference after CPP conditioning and **l** time difference, showing preference for the ON side on ChR2 but not on YFP animals. **n** Representative tracks for a ChR2 and a YFP animal during the RTPP. **o** Percentage and **p** difference of time spent on the ON and OFF sides, showing preference for the side associated with stimulation. Values are shown as mean ± s.e.m.*refers to difference between ChR2 stim+ and stim- levers, RM 2way ANOVA; ^refers to difference between ChR2 stim+ and YFP stim+ levers, RM 2way ANOVA. *$p < 0.05$; **$p < 0.01$; ***$p < 0.001$

increased motivational drive. There was an increased number of stim+ associated food pellets consumed in comparison to stim- associated food pellets in each progressive ratio session (Fig. 2h; RM 2way ANOVA, stim+ vs stim-: post hoc $t_{(34)} = 8.698$, $p < 0.0001$; ChR2 vs YFP stim+: $t_{(68)} = 3.06$, $p = 0.0063$).

**Optical activation of LDT-NAc inputs induces place preference.** We next evaluated if LDT-NAc stimulation per se was reinforcing in the conditioned place preference (CPP; non-contingent) and the real-time place preference (RTPP; contingent) tests, pairing a chamber of each apparatus with laser stimulation (ON chamber) (Fig. 2i, m, respectively).

Activating LDT-NAc terminals elicited place preference in the CPP, shown by the total time spent on the ON side (Fig. 2j; RM 2way ANOVA,ON vs OFF: post hoc $t_{(34)} = 5.584$, $p < 0.0001$) or by the increase in the ratio of preference between ON and OFF sides (Fig. 2k; RM 2-way ANOVA, ON vs OFF: post hoc $t_{(34)} = 8.366$, $p < 0.0001$). The difference of time spent between chambers was increased in the ChR2 group when compared to YFP group (Fig. 2l; $t$-test, $t_{(34)} = 4.104$, $p = 0.0002$).

Akin, in the RTPP, ChR2 animals also preferred the stimulus-associated ON chamber (Fig. 2n, o; RM 2way ANOVA, ON vs OFF: post hoc $t_{(34)} = 8.357$, $p < 0.0001$). Concordantly, the difference of time spent between chambers was increased in the ChR2 group when compared to control YFP group (Fig. 2p; $t$-test, $t_{(34)} = 6.288$, $p < 0.0001$).

Importantly, LDT-NAc stimulation did not affect locomotion nor appetite/food consumption (Supplementary Fig. 5f–h).

**Inhibition of LDT-NAc projections decreases preference and motivation.** We also performed optogenetic inhibition experiments using a similar strategy as before, but now injecting a cre-dependent hallorhodopsin in the LDT (AAV5-Ef1a-DIO-NpHR-eYFP-WPRE-pA; NpHR group) in combination with WGA-cre in the NAc. Electrophysiological measurements confirmed the functionality of this approach (Supplementary Fig. 6a–c). In sum, optical inhibition (4s constant yellow light) of LDT-NAc projections decreased NAc firing rate, with 56% of MSNs presenting decreased activity during the stimulation period (Supplementary Fig. 6d–h).

In the two-choice task, we observe an effect of session and group (Fig. 3a, b, RM 2way ANOVA, session: $F_{(7, 168)} = 26.34$, $p < 0.0001$; group: $F_{(3, 24)} = 20.04$, $p < 0.0001$). Optogenetic inhibition of LDT-NAc terminals decreased preference for the stim+ lever in NpHR animals (RM 2way ANOVA, stim+ vs stim-: post hoc $t_{(192)} = 11.33$, $p < 0.0001$). Control YFP group did not present preference for any lever, nor was the total amount of lever presses different between groups (Supplementary Fig. 5b).

In pellet extinction conditions, both groups decreased instrumental responding since the first session for both levers (Fig. 3c; RM 2way ANOVA, session: $F_{(4, 96)} = 218.2$, $p < 0.0001$). In laser extinction conditions, NpHR animals still manifested preference for stim- lever (Fig. 3d; RM 2way ANOVA, group: $F_{(3, 24)} = 42.68$, $p < 0.0001$; stim+ vs stim-: post hoc $t_{(96)} = 12.19$, $p < 0.0001$).

In the progressive ratio test (Fig. 3e), optical inhibition of LDT-NAc terminals decreased motivation, since NpHR animals showed less cumulative presses in the stim+ lever (Fig. 3f; RM 2way ANOVA, session: $F_{(5, 120)} = 132.7$, $p < 0.0001$, group: $F_{(3, 24)} = 4.952$, $p = 0.0081$; stim+ vs stim-: $t_{(144)} = 4.696$, $p < 0.0001$); and a robust decrease in the breakpoint (Fig. 3g; RM 2way ANOVA; $F_{(1, 12)} = 26.71$, $p = 0.0002$; $t_{(12)} = 8.006$, $p < 0.0001$), with no effect on the number of pellets consumed (Fig. 3h).

In the CPP or RTPP tests, we did not observe any significant effect of LDT-NAc optogenetic inhibition in NpHR animals in comparison to YFP animals (Fig. 3i–p).

**Differential recruitment of striatal populations in a task with manipulation of LDT-NAc inputs.** We next evaluated the activation pattern of different NAc neuronal populations during the PR task for either stim- lever or stim+ lever in YFP, ChR2 and NpHR animals. For this, we quantified the number of c-fos$^+$ cells and dopamine receptor D1 (D1R), dopamine receptor D2 (D2R), or choline acetyltransferase (ChAT) positive cells.

ChR2 stim+ group, in which animals worked for stim+ lever which was associated with LDT-NAc terminals optical stimulation, presented a significant increase in the number of c-fos$^+$/D1R$^+$ cells in comparison to ChR2 stim- group or YFP stim+ control group (Supplementary Fig. 7; 1way ANOVA, $F_{(5,21)} = 11.64$, $p < 0.0001$; stim+ vs stim-: post hoc $t(21) = 6.335$, $p < 0.0001$; ChR2 vs YFP stim+: post hoc $t(21) = 4.601$, $p = 0.0019$). Interestingly, this increase in cell activation was most evident in D1R$^+$ cells (Supplementary Fig. 7; 1way ANOVA, $F_{(5,21)} = 13.19$, $p < 0.0001$; stim+ vs stim-: post hoc $t(21) = 6.427$, $p < 0.0001$; ChR2 vs YFP stim+: post hoc $t(21) = 4.849$, $p = 0.0013$).

NpHR stim+ animals, in which animals worked for stim+ lever which was associated with LDT-NAc inputs optical inhibition, presented a reduction in c-fos$^+$/D1R$^+$ in comparison to NpHR stim- animals (1way ANOVA, stim+ vs stim-: post hoc $t(21) = 3.395$, $p = 0.0409$), though not different from control group. Conversely, they present increased number of c-fos$^+$/D2R$^+$ cells when compared to NpHR stim- animals (1way ANOVA, stim+ vs stim-: post hoc $t(21) = 3.79$, $p = 0.0161$;NpHR vs YFP stim+: post hoc $t(21) = 4.104$, $p = 0.0076$). No major differences were found in the number of recruited CINs (c-fos$^+$/ChAT$^+$) between groups.

Interestingly, we found that there was a positive correlation between the number of c-fos$^+$/D1R$^+$ cells and individual break-point (Person's correlation; $r = 6.187$, $p = 0.0006$). No behavioral correlation was found with the number of c-fos$^+$/D2R$^+$ (though there is a trend for negative correlation) nor with the number of c-fos$^+$/ChAT$^+$ cells (Supplementary Fig. 7).

**Effects of LDT-NAc cholinergic inputs optogenetic activation.** Since the majority of LDT-NAc projections were cholinergic, we decided to evaluate their contribution for the observed behavioral effects of non-selective LDT-NAc stimulation. We injected in the LDT of ChAT-cre mice an AAV5 expressing a cre-dependent ChR2 (ChAT-ChR2) or control eYFP-expressing virus (ChAT-YFP) (Fig. 4a).

The majority of YFP-expressing neurons in the LDT were cholinergic (Supplementary Fig. 8a, b). To test the functionality of this approach, we performed single-cell in vivo electrophysiological recordings. We found that there was a net increase in the firing rate of LDT during optogenetic stimulation of that region, with 77% of cells increasing their activity (Supplementary Fig. 8c, d, RM ANOVA, $F_{(1.315, 27.61)} = 20.75$, $p < 0.0001$, $n = 22$ cells).

In the NAc, LDT cholinergic terminals' optical stimulation evoked a net increase in NAc firing rate (Fig. 4b, RM ANOVA, $F_{(1.559, 71.71)} = 63.26$, $p < 0.0001$, $n = 47$ cells), inducing an excitatory response in 70% of recorded cells, whereas 9% presented an inhibitory response (Fig. 4c). The heatmap of firing rates of all recorded cells showed that most of cells increased firing rate during optical stimulation, but returned to baseline activity after (Fig. 4d). Analysis of distribution showed that firing rate significantly differed from a 10-s baseline window to the 4-s stimulus (Fig. 4e; Kolmogorov–Smirnov test, two tailed, $p < 0.001$). 83% of recorded pMSNs and 50% of recorded pFSs increased activity (Fig. 4f).

We then evaluated the role of cholinergic projections in mice behavior using the two-choice task as we have done for rats. There was an effect of session and group (Fig. 4g; RM 2way

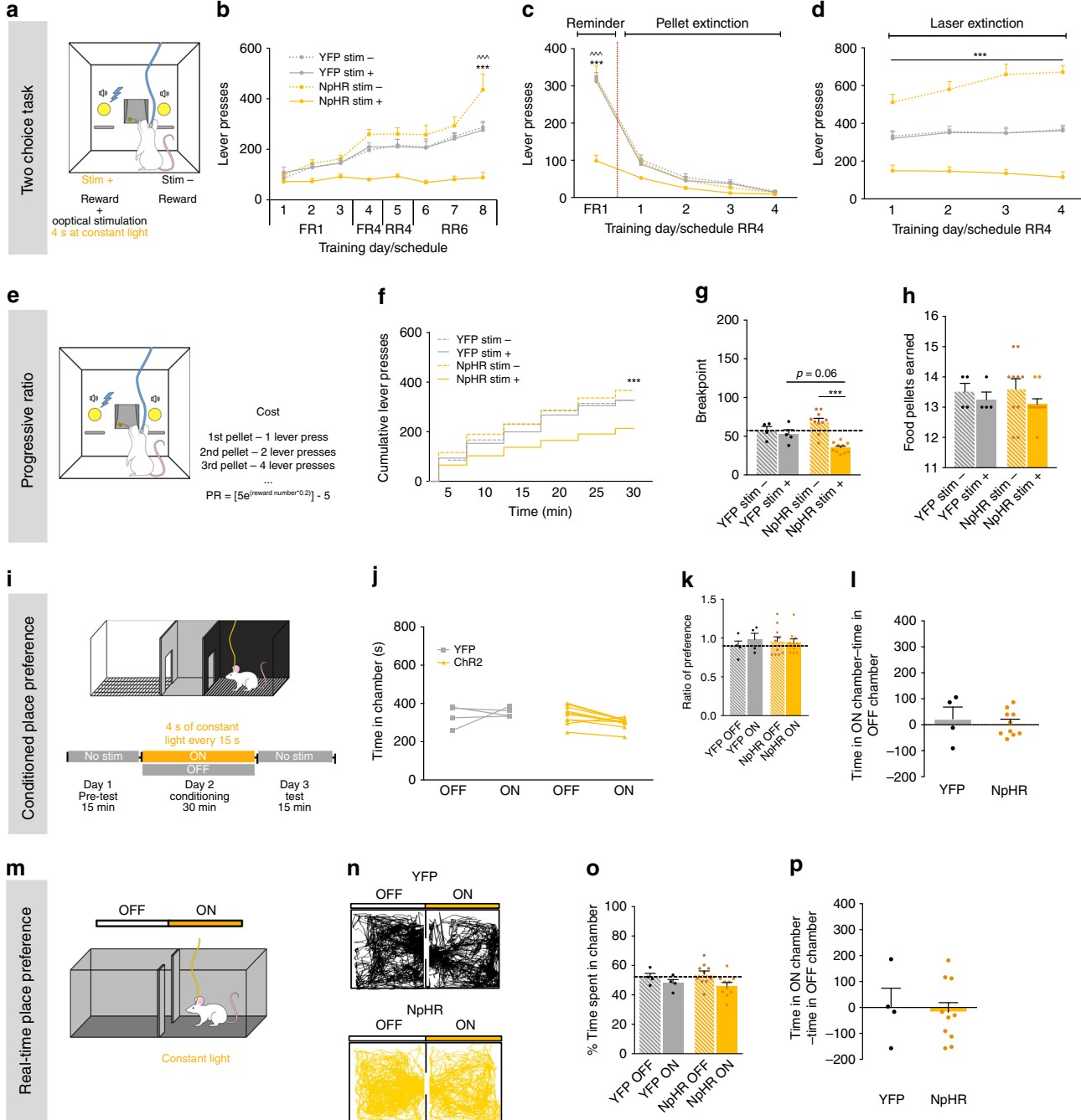

**Fig. 3** LDT-NAc optical inhibition decreases preference and motivation. **a** Schematic representation of the two-choice task. Pressing stim- lever yields one food pellet and pressing stim+ lever delivers a pellet + optical inhibition of LDT-NAc inputs (4s at constant light). **b** Time-course representation of the responses in NpHR ($n = 10$) and YFP ($n = 4$) rats. Optogenetic inhibition of LDT-NAc terminals shifts preference for the non-stimulated lever (stim-) in NpHR animals, but no preference is observed in YFP group. **c** In pellet extinction conditions, both groups decrease responses for both levers. **d** In laser extinction conditions, pressing either lever originates the delivery of a pellet, and stim+ no longer yields laser stimulation. NpHR animals still prefer stim-lever; YFP animals do not manifest preference. **e** Progressive ratio task. **f** Cumulative presses performed during the progressive ratio task show that NpHR animals press less on stim+ lever. **g** Decrease in breakpoint for stim+ lever in NpHR animals. **h** Number of food pellets earned during progressive ratio sessions. **i** CPP and **m** RTPP paradigms, in which one chamber is associated with NpHR-mediated inhibition of LDT-NAc projections (ON side). **j** Total time spent in the OFF and ON sides. **k** Ratio of preference after CPP conditioning and **l** time difference on the ON and OFF sides, showing no preference/ avoidance in any of the groups. **n** Representative tracks for an NpHR and a YFP animal during RTPP. **o** Percentage and **p** difference of time spent on the ON and OFF sides, showing no difference between groups. Values are shown as mean ± s.e.m. *refers to difference between NpHR stim+ and stim- levers, RM 2way ANOVA; ^refers to difference between NpHR stim+ and YFP stim+ levers, RM 2way ANOVA. *$p < 0.05$; **$p < 0.01$; ***$p < 0.001$

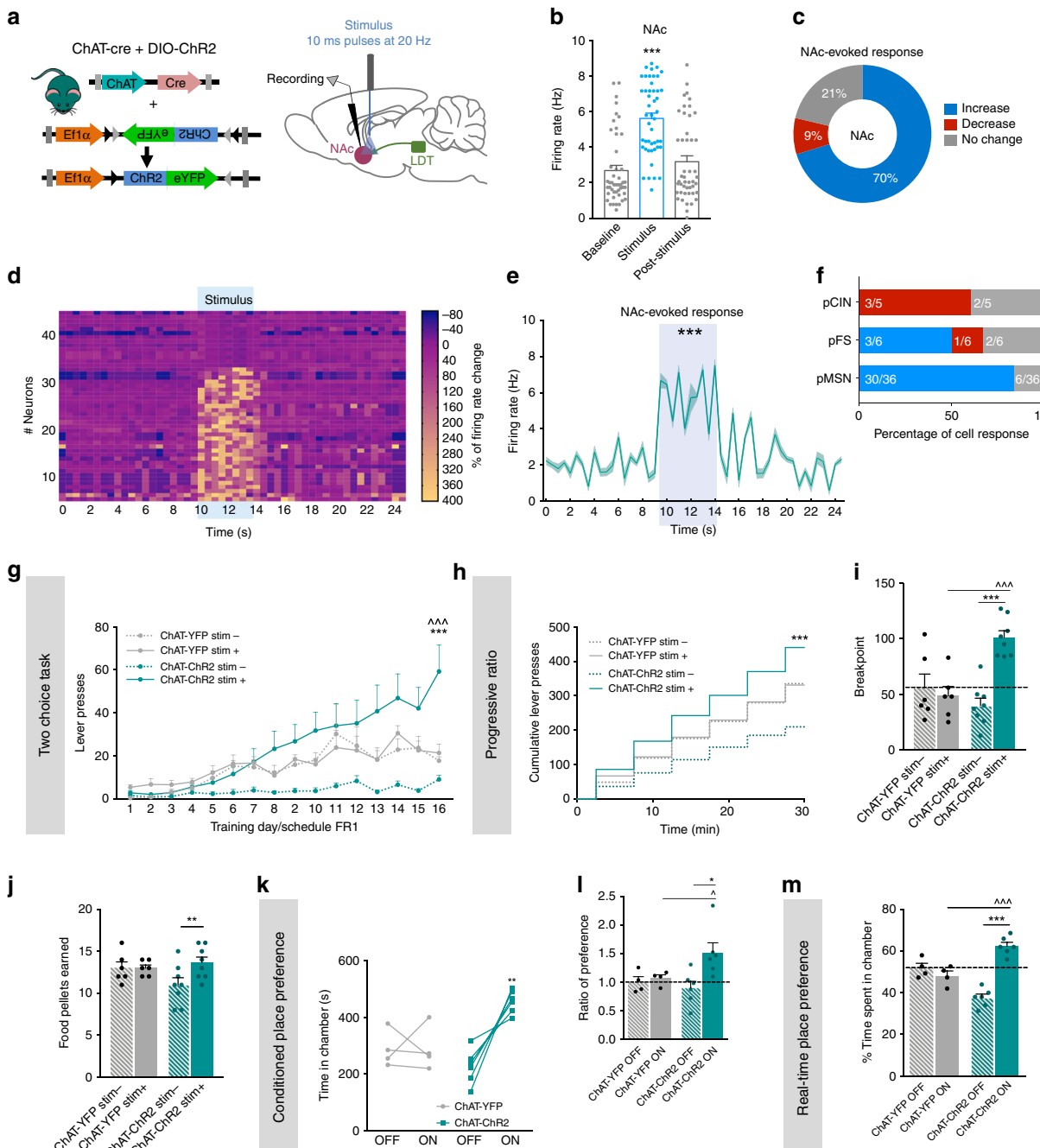

**Fig. 4** Activation of LDT-NAc cholinergic terminals increases motivation and induces preference. **a** Strategy used for optogenetic manipulation of LDT cholinergic terminals in the NAc. DIO-ChR2 was injected in the LDT of ChAT-cre mice, and optical stimulation was performed in the terminals in the NAc. **b** LDT cholinergic terminals stimulation increase NAc net firing rate ($n = 5$ animals; 47 cells; RM 1way ANOVA). **c** The majority of NAc cells show an increase in the firing rate during stimulation, 21% present no change and 9% decrease activity. **d** Heatmap representation of percentages of cell responses in the NAc when LDT cholinergic terminals are stimulated. **e** Average distribution firing rate of NAc neurons showing an increase in activity during stimulation period (KS test). **f** Percentage of responses of each NAc cell type to cholinergic LDT terminal activation. **g** Time-course representation of the responses in the two-choice task of ChAT-ChR2 ($n = 8$) and ChAT-eYFP ($n = 6$) mice. Activation of LDT-NAc cholinergic terminals enhances responses for stim+ lever in ChAT-ChR2 animals, but not in control ChAT-YFP group. **h** Cumulative presses performed during the progressive ratio task show that ChAT-ChR2 animals press more on stim+ lever. **i** Increased breakpoint for stim+ lever in ChAT-ChR2 animals, indicative of enhanced motivation. **j** Number of pellets consumed during progressive ratio sessions. **k–l** In the CPP, ChAT-ChR2 ($n = 6$, two animals lost cannula) animals prefer the ON chamber, i.e., the one associated with NAc-LDT cholinergic stimulation, whereas no preference is seen in control group ($n=4$, two animals lost cannula). **m** In the RTPP test, ChAT-ChR2 animals also prefer the ON chamber. Values are shown as mean ± s.e.m. *refers to difference between ChAT-ChR2 stim+ and stim- lever, RM 2way ANOVA; ˆrefers to difference between ChAT-ChR2 stim+ and ChAT-YFP stim+ lever, RM 2way ANOVA. *$p < 0.05$; **$p < 0.01$; ***$p < 0.001$

ANOVA, session: $F_{(15, 360)} = 17.1$, $p < 0.0001$; group: $F_{(3, 24)} = 6.212$, $p = 0.0028$). ChAT-ChR2 mice progressively discriminate and prefer the stim+ over the stim- lever (Fig. 4g; RM 2way ANOVA, stim+ vs stim-: post hoc $t_{(384)} = 7.059$, $p < 0.0001$). No lever preference was observed in ChAT-YFP animals.

In pellet extinction conditions, all groups decrease instrumental responding as soon as the first session, extinguishing response (Supplementary Fig. 8e; RM 2way ANOVA, session: $F_{(10, 240)} = 95.37$, $p < 0.0001$).

In laser extinction conditions, ChAT-ChR2 mice still displayed preference for the stim+ in comparison to stim- lever (Supplementary Fig. 8f; RM 2way ANOVA, stim+ vs stim-: post hoc $t_{24} = 11.37$, $p < 0.0001$); while ChAT-YFP control animals do not show any preference. To further rule out confounding factors such as instrumental habituation, we confirmed behavioral flexibility in a single-session reversal paradigm (Supplementary Fig. 8g).

In the progressive ratio test, both group and session had a significant effect in cumulative lever presses (Fig. 4h; RM 2way ANOVA, group: $F_{(3, 24)} = 2.694$, $p = 0.0686$; session: $F_{(5, 120)} = 94.77$, $p < 0.0001$). ChAT-ChR2 animals presented a stable increase in cumulative presses in the stim+ lever (RM 2way ANOVA, stim+ vs stim-: post hoc $t_{(144)} = 1.844$, $p = 0.0003$); displaying a marked augmentation in the stim+ breakpoint (Fig. 4i; RM 2way ANOVA, stim+ vs stim-: post hoc $t_{(12)} = 8.871$, $p < 0.0001$), indicating increased motivation to work in that lever. These animals also consumed more pellets in stim+ lever (Fig. 4j; RM 2way ANOVA, stim+ vs stim-: post hoc $t_{(12)} = 4.706$, $p = 0.001$).

Akin to rat data, optical activation of LDT-NAc cholinergic terminals recapitulated the enhanced preference for the ON chamber in both the CPP (Fig. 4k; RM 2way ANOVA, ON vs OFF: post hoc $t_{(8)} = 5.427$, $p = 0.0013$; ratio of preference: Fig. 4l; RM 2way ANOVA, ON vs OFF: post hoc $t_{(8)} = 2.843$, $p = 0.0434$) and RTPP tests (Fig. 4m; RM 2way ANOVA, ON vs OFF: post hoc $t_{(16)} = 9.507$, $p < 0.0001$).

In sum, optical activation of LDT-NAc cholinergic projections shifts preference, enhances motivation and induces place preference.

**Effects of LDT-NAc glutamatergic inputs optogenetic activation.** Considering the different nature of LDT-NAc projections, we also assessed the role of glutamatergic inputs in behavior. We injected in the LDT of VGluT-cre mice, a cre-dependent ChR2 (or YFP) for glutamatergic manipulation (Fig. 5a).

LDT glutamatergic terminals optical stimulation evoked a net increase in NAc firing rate (Fig. 5b, RM 1way ANOVA, $F_{(1.972, 90.73)} = 31.08$, $p < 0.0001$; $n = 47$ cells); inducing an excitatory response in 45% of recorded cells, whereas 23% presented an inhibitory response (Fig. 5c). The heatmap of firing rates of all recorded cells showed that most of cells increased firing rate during optical stimulation (Fig. 5d). Firing rate significantly differed from a 10-s baseline window to the 4-s stimulus (Fig. 5e; Kolmogorov–Smirnov test two tailed, $p = 0.0434$) and around half of MSNs display an increase in the firing rate (Fig. 5f).

In the two-choice task, optical activation of LDT-NAc glutamatergic terminals (80 10 ms pulses at 20 Hz) increased presses in stim+ lever, but this effect was not so evident as for cholinergic projections (Fig. 5g; RM 2way ANOVA, session: $F_{(15, 450)} = 16.66$, $p < 0.0001$; group: $F_{(3, 30)} = 8.34$, $p = 0.0004$).

In pellet extinction conditions, there was no difference in lever presses between stim+ and stim- in both groups (Supplementary Fig. 9a). In laser extinction conditions, VGlut-ChR2 mice decrease preference for the stim+ throughout sessions (Supplementary Fig. 9b; RM 2way ANOVA, session: $F_{(3, 90)} = 4.908$, $p = 0.0033$; stim+ vs stim-: post hoc $t_{(120)} = 1.763$, $p = 0.4830$).

Behavioral flexibility was confirmed in a single-session reversal paradigm (Supplementary Fig. 9c).

In the progressive ratio test, VGlut-ChR2 animals presented increased presses in stim+ lever in comparison to stim- (Fig. 5h; RM 2way ANOVA, session: $F_{(5, 150)} = 132.8$, $p < 0.0001$; stim+ vs stim-: post hoc $t_{(180)} = 3.042$, $p = 0.0162$). VGluT-ChR2 animals also presented a higher breakpoint for stim+ in comparison to stim- lever (Fig. 5i; RM 2way ANOVA, stim+ vs stim-: post hoc $t_{(15)} = 3.608$, $p = 0.0052$), with no differences in the number of pellets consumed (Fig. 5j). However, this enhanced motivation for pressing the stim+ lever of VGluT-ChR2 mice was not significantly different from YFP group.

Specific activation of glutamatergic LDT-NAc projections was not sufficient to elicit place preference in the CPP (Fig. 5k, l). Surprisingly, in the RTPP test, VGlut-ChR2 group presented avoidance to the ON chamber (Fig. 5m; RM 2way ANOVA, side chamber: $F_{(1, 22)} = 7.854$, $p = 0.0104$; ON vs OFF: post hoc $t_{(22)} = 3.512$, $p = 0.0118$).

**Effects of LDT-NAc GABAergic inputs optogenetic activation.** Next, we evaluated the role of LDT-NAc GABAergic inputs in behavior. We injected in the LDT of VGAT-cre mice, a cre-dependent ChR2 (or YFP) for GABAergic inputs manipulation (Fig. 6a).

LDT GABAergic terminals optical stimulation evoked a net decrease in NAc firing rate (Fig. 6b, RM 1way ANOVA, $F_{(1.413, 73.49)} = 17.41$, $p < 0.0001$; $n = 53$ cells); inducing an inhibitory response in 62% of recorded cells, whereas 11% presented an excitatory response (Fig. 6c). The heatmap of firing rates of all recorded cells showed that most of cells decreased firing rate during optical stimulation, but returned to baseline activity after (Fig. 6d). Firing rate was significantly decreased from a 10-s baseline window to the 4-s stimulus (Fig. 6e; Kolmogorov–Smirnov test two tailed, $p = 0.0003$) and most MSNs recorded showed a decrease in the firing rate (Fig. 6f).

In the two-choice task, optical activation of these LDT-NAc GABAergic projections (80 10 ms pulses at 20 Hz) induced a significant effect of group and session (Fig. 6g; RM 2way ANOVA, session: $F_{(15, 450)} = 7.976$, $p < 0.0001$; group: $F_{(3, 30)} = 6.062$, $p = 0.0024$). LDT-NAc GABAergic activation shifted preference for stim- lever in VGAT-ChR2 animals (RM 2way ANOVA, stim+ vs stim-: post hoc $t_{(480)} = 4.841$, $p < 0.0001$). YFP group displayed no preference for any lever, as expected.

In pellet extinction conditions, there was no difference in lever presses between stim+ and stim- in both groups (Supplementary Fig. 10a). In laser extinction conditions, VGAT-ChR2 mice still prefer stim- lever until the last session (Supplementary Fig. 10b; RM 2way ANOVA, session: $F_{(3, 90)} = 3.023$, $p = 0.0337$; stim+ vs stim-: post hoc $t_{(120)} = 1.782$, $p = 0.4634$). Behavioral flexibility was confirmed in a single-session reversal paradigm (Supplementary Fig. 10c).

In the progressive ratio task, VGAT-ChR2 displayed more lever presses in stim- than in stim+ lever (Fig. 6h; RM 2way ANOVA, group: $F_{(3,30)} = 2.987$, $p = 0.0467$, session: $F_{(5, 150)} = 151.7$, $p < 0.0001$; stim+ vs stim-: post hoc $t_{(180)} = 4.336$, $p < 0.0001$), originating decreased motivation to work for the laser-paired reward (Fig. 6i; RM 2way ANOVA, stim + vs stim -: post hoc $t_{(15)} = 4.124$, $p = 0.0018$); and also in comparison to YFP group (VGAT-ChR2 vs VGAT-YFP stim+: t-test, $t_{(30)} = 2.706$, $p = 0.0222$). VGAT-ChR2 animals consumed less pellets in the stim- lever session (Fig. 6j; RM 2way ANOVA, stim + vs stim -: post hoc $t_{(15)} = 2.726$, $p = 0.0312$).

Of note, activation of LDT-NAc GABAergic inputs was not able to induce place preference nor aversion in the CPP or RTPP paradigms (Fig. 6k–m).

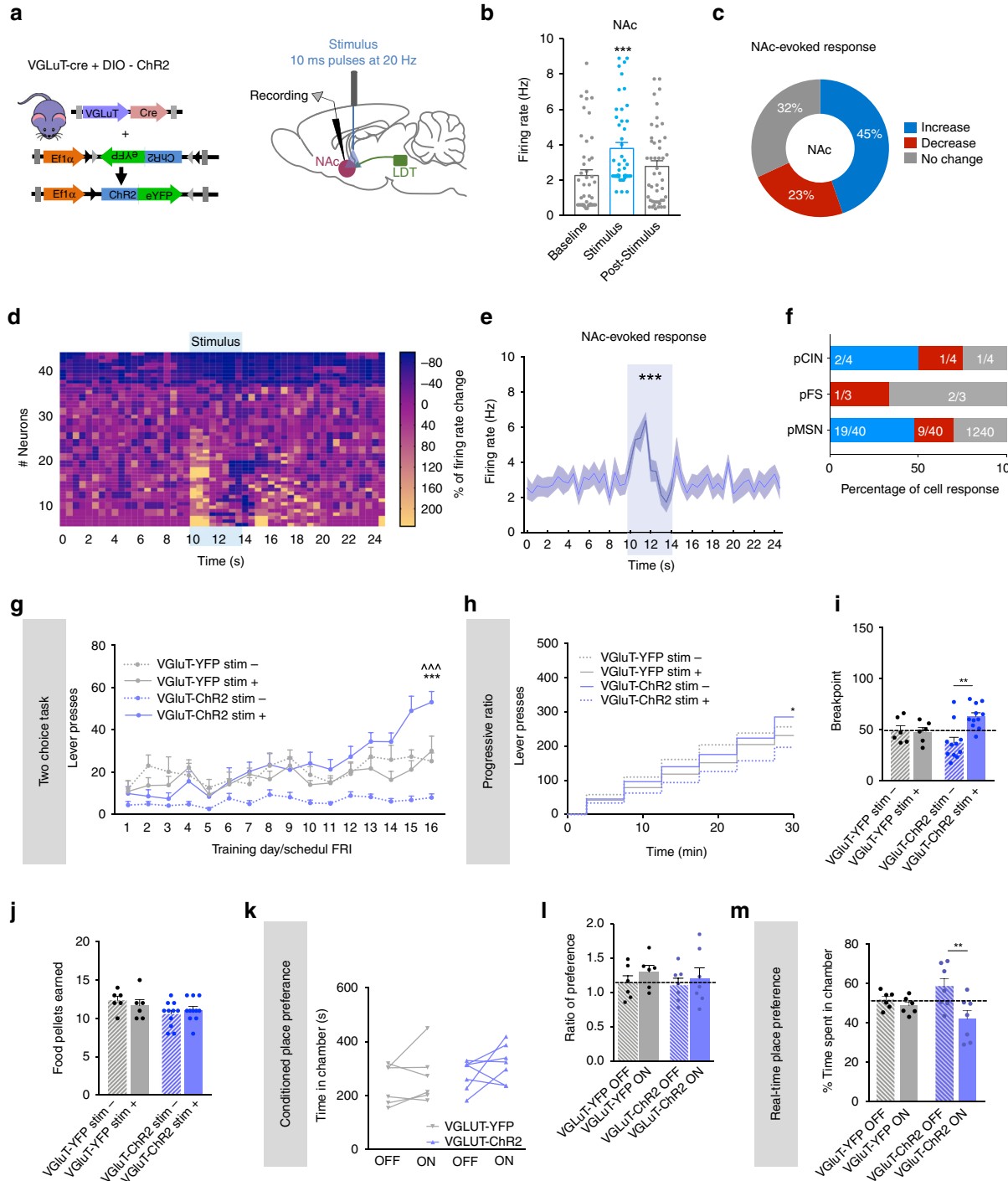

**Fig. 5** Optogenetic activation of LDT-NAc glutamatergic inputs during reward-related behaviors. **a** Strategy used for optogenetic manipulation of LDT glutamatergic terminals in the NAc. DIO-ChR2 was injected in the LDT of VGluT-cre mice, and optical stimulation was performed in the terminals in the NAc. **b** Stimulation of glutamatergic terminals increase NAc net firing rate ($n = 4$ animals; 42 cells; RM 1way ANOVA). **c** 45% of NAc cells show an increase in the firing rate during stimulation, 32% present no change and 23% decrease activity. **d** Heatmap representation of percentages of cell responses in the NAc upon stimulation of glutamatergic terminals. **e** Average firing rate of NAc neurons showing an increase in activity from the baseline during stimulation (KS test). **f** Percentage of responses of each NAc cell type to glutamatergic LDT terminal activation. **g** Time-course representation of the responses in the two-choice task of VGluT-ChR2 ($n = 11$) and VGluT-YFP ($n = 6$) mice on stim+ and stim- lever. VGluT-ChR2 animals prefer the stim+ over stim- lever; no preference is observed in VGluT-YFP animals. **h** Cumulative presses performed during the progressive ratio task, showing increased presses in stim+ lever in VGluT-ChR2 animals. **i** Breakpoint is increased on VGluT-ChR2 animals for the stim+ in comparison to stim- lever, though not statistically different from control group. **j** Number of pellets consumed during progressive ratio sessions. **k** Total time spent in the ON and OFF chambers in the CPP test, showing no preference. **l** Ratio of preference between ON and OFF chambers. **m** Percentage of time spent in each side of the RTPP test, showing decreased time spent in the ON chamber in VGlut-ChR2 animals ($n = 7$, four animals lost canula) but not in VGlut-YFP group. Values are shown as mean ± s.e.m. *refers to difference between VGluT-ChR2 stim+ and stim- lever, RM 2way ANOVA; ^refers to difference between VGluT-ChR2 stim+ and VGluT-YFP stim+ lever, RM 2way ANOVA. *$p < 0.05$; **$p < 0.01$; ***$p < 0.001$

## Discussion

The LDT has been linked with locomotion, sleep and, lately, with reward-related behaviors[20,21]. Initially, the contribution of LTD for reward was proposed based on its modulatory role of VTA activity, ultimately influencing dopamine release in the NAc[11,13,22]. Indeed, optogenetic stimulation of LDT-VTA neurons enhances conditioned place preference[3] and operant response in rats[6,15]. However, the discovery that the LDT also sends inputs to the NAc[16,23] suggests that this brain region can also modulate accumbal activity and influence reward behaviors in a VTA-independent manner. In agreement, here we show that the LDT sends inputs of different neurotransmitter nature to the NAc, and that optogenetic modulation of LDT (cholinergic) terminals affects different dimensions of reward-related behaviors.

Stimulation of all LDT-NAc terminals amplifies a reward previously paired with laser stimulation over an otherwise identical reward. In laser extinction conditions, animals still display preference for the stim+ lever, showing that the associated reward/lever gained increased value for the animal. Conversely, optogenetic inhibition of LDT-NAc terminals shifts preference for stim- lever. Altogether, these results indicate that LDT-NAc activation/inhibition is able to add/decrease the value of a laser-associated reward, respectively.

Interestingly, preference for the stim+ lever was completely abolished in LDT-NAc stimulated animals if the reward was omitted (pellet extinction conditions). These findings are similar to another study, in which optogenetic stimulation of central amygdala increased preference for a specific lever but only if paired with an external food reward[24].

Interestingly, in laser extinction conditions (two levers yield similar reward), ChR2 animals still present a robust preference for the stim+ lever though no stimulation is given. Of notice, ChR2 animals even slightly increase the number of presses in this lever (conversely, NpHR animals increase presses in stim- lever). One can hypothesize that since these animals were previously exposed to pellet extinction conditions, now, when the reward is present again, they readjust the pattern of lever presses in the previously preferred (reward-associated) lever. In other words, it can reflect enhanced salience of the preferred lever when the reward is available again. Unfortunately, in this present study we did not register the temporal pattern of lever press, which would be important to better understand these findings.

Because animals still preferred the stim+ lever in the absence of laser stimulation, one can also argue that LDT-NAc optogenetic activation transformed the value of its associated external reward, rather than representing an internal reinforcement state. However, this finding appears paradoxical in the light of our findings showing that stimulation of LDT-NAc inputs is also reinforcing per se, since it induces a robust place preference in the CPP and RTPP paradigms. This is explained by the context in which laser stimulation is paired with: in the two-choice task, animals are food deprived and their main objective is to obtain a food pellet, in a way that stimulation adds value to a paired representation of a physiological need of the animal; in contrast, in the place preference tests, animals are satiated and have no physiological need to fulfill, except for their natural curiosity in exploring new environments. This highlights the complexity of reward signaling, and the importance of the context and associative learning in the process. Other studies have also originated puzzling effects, in which a rewarding stimulus can act as reinforcer in one context but not in others[25].

In addition to this increase in reward saliency, we found that LDT-NAc stimulation amplified motivation towards the laser-paired reward, as shown by the 115% increase of the breakpoint. One could argue that these effects were due to an increase in

"liking" the reward[26] however, stimulation during free feeding behavior did not induce any consumption differences for chow or palatable food (Supplementary Fig. 5).

Considering the available data regarding NAc role in reinforcement, one could hypothesize that any excitatory projection to the NAc could lead to reward/positive reinforcement and enhance motivation, but most likely it depends on where in the NAc and to which type of striatal neuron it projects preferentially to. For example, rodents will self-stimulate for amygdala-NAc glutamatergic inputs, yet, they do not self-stimulate for glutamatergic fibers from the medial prefrontal cortex to the NAc (which also evoked an excitatory synaptic response)[27–29]. In line with our data, cholinergic and glutamatergic LDT-NAc inputs also drive a net excitatory response in the NAc but originate different behavioral outcomes (see below).

We found that 50% of LDT-NAc projections were cholinergic, with additional contribution of glutamatergic and GABAergic neurons. These cholinergic inputs provide a predominantly excitatory input to NAc MSNs, although it is not clear which NAc neurons they preferentially innervate. Our data suggests that LDT-NAc optical activation effects may be mostly mediated by D1-MSNs, since these neurons were more recruited in animals working for stim+ versus those working for stim-lever. Interestingly, there was also a positive correlation between the number of activated D1R+ cells in the NAc and individual breakpoint, whereas no significant effect was observed for D2-MSNs (c-fos+/D2R+) nor for cholinergic interneurons (c-fos+/ChAT+). It is important to refer that while this manuscript was in revision, one study showed that in dorsal striatum, PPT/LDT cholinergic inputs preferentially innervate and activate cholinergic interneurons, although part of these inputs also innervate dorsal striatal MSNs[30]. This highlights the complexity of the cholinergic modulation of striatal activity and function, and emphasizes the need for additional studies to dissect how mesopontine nuclei coordinate striatal and midbrain (VTA and substantia nigra) activity.

Optical stimulation of LDT-NAc cholinergic inputs recapitulated the shift in preference for the laser-associated lever, but again, only when paired with the external food reward. LDT-NAc cholinergic terminals stimulation also elicited place preference, showing that it also conveys reinforcing properties. So, one can hypothesize that LDT-NAc cholinergic projections account for most of the positive effects in behavior observed in our rat experiments, in which all types of LDT projections were activated. It is known that acetylcholine release in the striatum can have a plethora of actions, as it has a differential effect in MSNs or interneurons and it influences neurotransmitter release via presynaptic mechanisms[31,32]. However, one caveat of most published studies was the assumption that the only source of striatal acetylcholine was originated from cholinergic interneurons. Considering our data, one can hypothesize that M2 and M4 receptors in cholinergic interneurons are not solely autoreceptors[33–35], but can also decode signals from LDT cholinergic inputs. In addition, LDT-acetylcholine can regulate corticostriatal glutamatergic or VTA dopaminergic release via presynaptic acetylcholine receptors for example[31,32,35].

Although LDT cholinergic inputs appear to be sufficient to recapitulate the positive/rewarding effects of non-selective LDT-NAc stimulation, one cannot disregard the contribution of other neuronal types. Activation of LDT-NAc glutamatergic projections also enhanced preference for the laser-associated reward, and increased motivational drive, but this effect was not so evident as for cholinergic inputs modulation. Moreover, to our surprise, stimulation of LDT-NAc glutamatergic inputs had no effect in the CPP but induced avoidance in the RTPP test. Although the circuit is different, Yoo et al.[36] showed that, contrary to brief stimulation

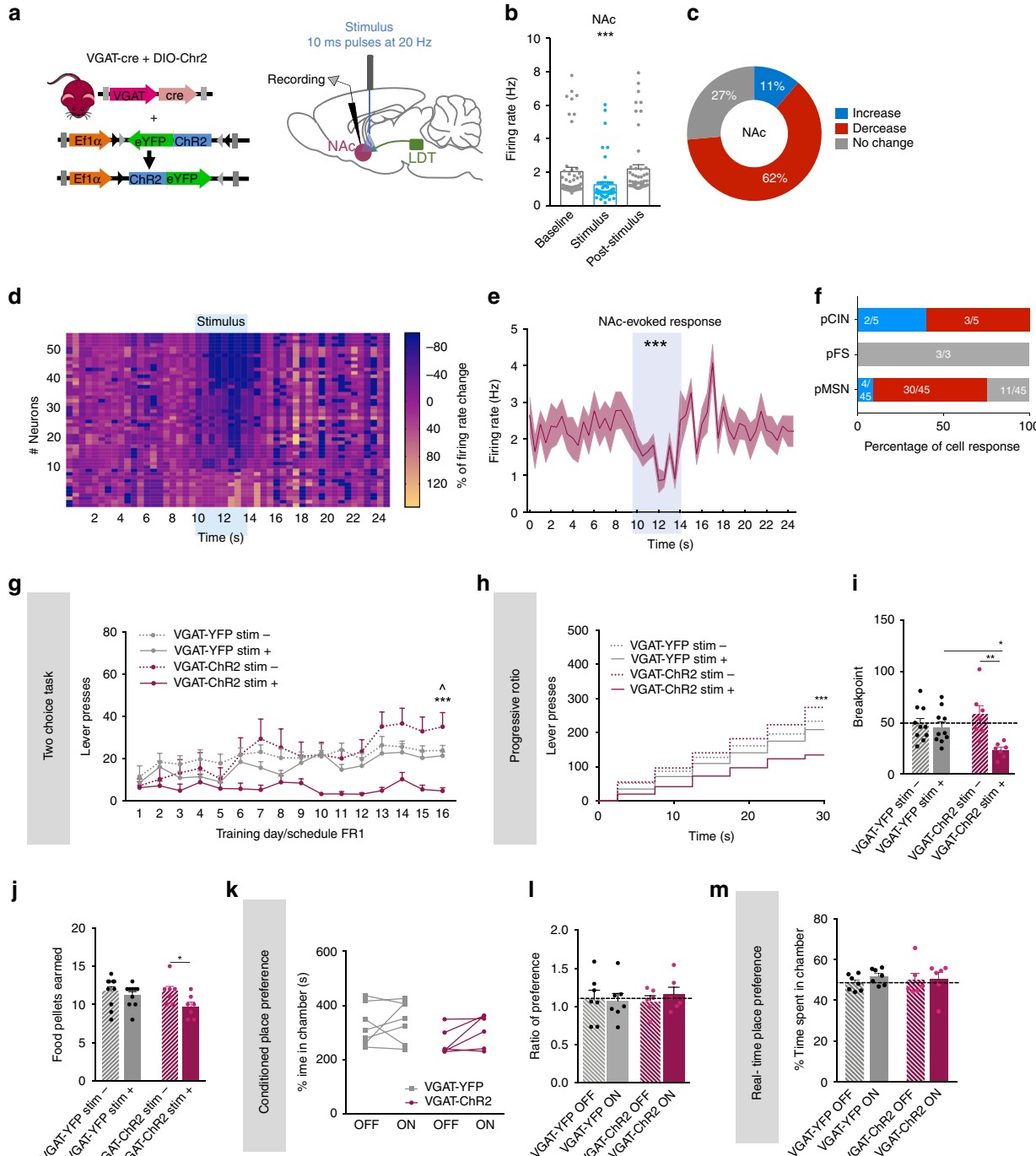

**Fig. 6** Selective activation of LDT-NAc GABAergic inputs during reward-related behaviors. **a** Strategy used for optogenetic manipulation of LDT glutamatergic terminals in the NAc. DIO-ChR2 was injected in the LDT of VGAT-cre mice, and optical stimulation was performed in the terminals in the NAc. **b** Stimulation of GABAergic terminals decrease NAc net firing rate ($n = 6$ animals; 53 cells; RM 1way ANOVA). **c** The majority of NAc cells decreased the firing rate during stimulation, 27% presented no change and 11% increased activity. **d** Heatmap representation of percentages cell responses in the NAc upon stimulation of GABAergic terminals. **e** Average firing rate of NAc neurons showing a decrease in activity from the baseline during optical inhibition (KS test). **f** Percentage of responses of each NAc cell type to GABAergic LDT terminal activation. **g** Time-course representation of the responses during the two-choice task of VGAT-ChR2 ($n = 7$) and VGAT-YFP ($n = 10$) mice on stim+ and stim- lever. VGAT-ChR2 animals prefer the stim- over stim+ lever; no preference is observed in VGAT-YFP animals. **h** Cumulative presses performed during the progressive ratio task, showing increased presses in stim- lever in VGAT-ChR2 animals. **i** Breakpoint for stim+ lever is decreased in VGAT-ChR2 animals, indicative of decreased motivation for the laser-paired reward. **j** Number of pellets earned during progressive ratio sessions. **k** Total time spent in ON and OFF chambers in the CPP ($n_{VGAT-ChR2} = 6$, one animal lost cannula; $n_{VGAT-YFP} = 7$, four animals lost cannula). **l** Ratio of preference between ON and OFF chambers in the CPP test, showing no differences between groups. **m** Percentage of time spent in each side of the RTPP, showing no differences between groups. Values are shown as mean ± s.e.m. *refers to difference between VGAT-ChR2 stim+ and stim- lever, RM 2way ANOVA; ^refers to difference between VGAT-ChR2 stim+ and VGAT-YFP stim+ lever; RM 2way ANOVA. *$p < 0.05$; **$p < 0.01$; ***$p < 0.001$

of VTA-NAc glutamatergic terminals which induced positive reinforcement, sustained glutamatergic signals convey aversion, probably by acting on GABAergic interneurons in the NAc[37,38]. In the RTPP, animals spent an average of 1 min per entry in the ON chamber, so, the prolonged activation of LDT-NAc glutamatergic inputs could also lead to a similar aversive effect.

Lastly, we show that activation of LDT-NAc GABAergic inputs decreased the value of a laser-paired reward, decreased motivation, though it did not induce preference nor aversion in the CPP or RTPP tests. These findings require deeper investigation, especially in the light of data suggesting that NAc GABA mediates motivated/affective behavior that is bivalently organized along a rostrocaudal gradient[39].

In summary, we show that LDT-NAc inputs mainly drive an excitatory response in the NAc, and play a pro-rewarding role. Of notice, this effect may be region specific, since a recent work has shown that activation of cholinergic PPT/LDT inputs can also decrease dorsal striatum MSN firing rate[30]. Most of the LDT-NAc projections are cholinergic, but there is a subset of glutamatergic and GABAergic inputs. LDT-NAc cholinergic, and less prominently glutamatergic, inputs convey rewarding signals, whereas GABAergic inputs appear not to. Our findings provide functional evidence regarding LDT-NAc inputs, and emphasize the importance of additional studies to dissect how the LDT integrates reward information and signals through the NAc directly (and via VTA) to drive reward-related behaviors.

## Methods

**Animals and treatments**. Wistar Han rats, aged 2 months at the beginning of experiments, were housed under standard laboratory conditions (light/dark cycle of 12/12 h; 22 °C); food and water ad libitum.

Male and female C57/Bl6 transgenic mice (age of 2 months at the beginning of the experiments) were housed at weaning in groups of 3–5 animals per cage, under standard laboratory conditions (light/dark cycle of 12/12 h; 22 °C); food and water ad libitum. The progeny produced by homozygous VGAT-cre (Vgat-IRES-Cre: Slc32a1$^{tm2(cre)Lowl/J}$, The Jackson Laboratory) and VGluT-cre (VGluT2-IRES-Cre: Slc17a6$^{tm2(cre)Lowl/J}$, The Jackson Laboratory) transgenic mice were used. The progeny produced by mating ChAT-cre (B6;129S6-Chat$^{tm2(cre)Lowl}$/J/ChAT-IRES-Cre, #006410, The Jackson Laboratory) heterozygous transgenic male mice with wild-type C57/Bl6 females were genotyped at weaning by PCR fragment analysis.

All behavioral experiments were performed during the light period of the light/dark cycle. Health monitoring was performed according to FELASA guidelines, confirming the Specified Pathogen Free health status of sentinel animals maintained in the same animal room. All procedures were conducted in accordance with European Regulations (European Union Directive 2010/63/EU).

Animal facilities and the people directly involved in animal experiments were certified by the Portuguese regulatory entity—Direção Geral de Alimentação e Veterinária (DGAV). All the experiments were approved by the Ethics Committee of the University of Minho (SECVS protocol #107/2015). The experiments were also authorized by the national competent entity DGAV (#19074).

**Genotyping**. DNA was isolated from tail biopsy using the Citogene DNA isolation kit (Citomed). In a single PCR genotyping tube, the primers CRE F (5′-AGCCT GTTTTGCACGTTCACC-3′) and Cre R (5′-GGTTTCCCGCAGAACCTGAA-3′) were used to amplify the cre transgene. An internal control gene (metallothionein 3) was used in the PCR (MT3_F (5′-GGTCCTCACTGGCAGCAGCTGCA-3′) and MT3_2 (5′-CCTAGCACCCACCCAAAGAGCTG-3′). Heterozygous mice were discriminated from the wild-type mice by the presence of two amplified DNA products corresponding to the transgene and the internal control gene. Gels were visualized with GEL DOC EZ imager (Bio-Rad, Hercules, CA, USA) and analyzed with the Image Lab 4.1 (Bio-Rad).

**Optogenetics—constructs**. AAV5–EF1a–WGA–Cre–mCherry, AAV5–EF1a–DIO–hChR2–YFP, AAV5–EF1a–DIO–eNpHR 3.0–YFP and AAV5–EF1a–DIO–YFP were obtained directly from the Gene Therapy Center Vector Core (UNC) center. AAV5 vector titers were 2.1-6.6X10$^{12}$ virus molecules/ml.

**Surgery and cannula implantation**. Rats designated for behavioral experiments were anesthetized with 75 mg kg$^{-1}$ ketamine (Imalgene, Merial) plus 0.5 mg kg$^{-1}$ medetomidine (Dorbene, Cymedica). One µl of AAV5–EF1a–WGA–Cre–mCherry was unilaterally injected into the NAc (coordinates from bregma, according to Paxinos and Watson:[40] +1.5 mm anteroposterior (AP), +0.9 mm mediolateral (ML), and

−6.5 mm dorsoventral (DV)) and 1 µl of AAV5–EF1a–DIO–hChR2–YFP was injected in the LDT (coordinates from bregma: −8.5 mm AP, +0.9 mm ML, and −6.5 mm DV) (ChR2 group). Another group of animals was injected in the LDT with 1 µl of AAV5–EF1a–DIO–eNpHR 3.0–YFP (NpHR). Control group was injected with AAV5–EF1a–DIO–YFP in the LDT and WGA–Cre in the NAc. An additional set of experiments were performed to demonstrate the specificity of this approach, by injecting the virus in LDT adjacent areas—periaqueductal gray and 4th ventricle (Supplementary Fig. 11–12).

Rats were then implanted with an optic fiber (200 µm core fiber optic; Thorlabs, NJ, USA) with 2.5 mm stainless steel ferrule (Thorlabs, NJ, USA) 0.1 mm above the injection coordinates for the NAc. Ferrules were secured to the skull using 2.4 mm screws (Bilaney, Germany) and dental cement (C&B kit, Sun Medical). Rats were removed from the stereotaxic frame and sutured. Anesthesia was reverted by administration of atipamezole (1 mg/kg). After surgery animals were given anti-inflammatory (Carprofeno, 5 mg/kg) for one day, analgesic (butorphanol, 5 mg/kg) for 3 days, and were let to fully recover before initiation of behavior. Optic fiber placement was confirmed for all animals after behavioral experiments. Animals that were assigned for electrophysiological experiments were not implanted with an optic fiber.

Mice designated behavioral experiments were anaesthetized with 75 mg kg$^{-1}$ ketamine (Imalgene, Merial) plus 1 mg kg$^{-1}$ medetomidine (Dorbene, Cymedica). AAV5–EF1a–DIO–hChR2–YFP (or AAV5–EF1a–DIO–YFP) virus (500 nl) was unilaterally injected into the LDT (coordinates from bregma, according to Paxinos and Franklin:[41] −5 mm AP, +0.5 mm ML, and −3.0 mm DV) using an 30-gauge needle Hamilton syringe (Hamilton Company, Switzerland), at a rate of 100 nl min$^{-1}$; the syringe was left in place for 5 min to allow diffusion. Mice were implanted with an optic fiber (200 µm core fiber optic; Thorlabs) with 2.5 mm stainless steel ferrule (Thorlabs) in the NAc (coordinates from bregma): +1.2 mm AP, +0.8 mm ML, and −3.8 mm DV, and secured to the skull using dental cement (C&B kit, Sun Medical). Mice were removed from the stereotaxic frame, sutured and left to recover for two weeks before initiation of the behavioral protocols. All animals were treated 30 min before and 6 h after surgery with an analgesic—buprenorphine at 0.05 mg kg$^{-1}$ (Bupaq, Richterpharma).

**In vivo electrophysiology recordings**. Four weeks after injection of the virus, animals were submitted to a stereotaxic surgery for the placement of the optic fiber and recording electrodes, following anatomical coordinates. Animals were anesthetized with urethane (1.75 g Kg$^{-1}$, Sigma). The total dose was administered in three separate intra peritoneal injections, 30 min apart. Body temperature was maintained at ~37 °C with a homeothermic heat pad system (DC temperature controller, FHC, ME, USA). Adequate anesthesia was confirmed by observation of general muscle tone, by assessing withdrawal responses to noxious pinching and by whiskers movement.

Stimulating and recording electrodes were placed in the following coordinates: for rats—LDT: −8.5 mm AP, 0.9 mm ML, −6.3 to −7.9 mm DV; NAc: +1.5 mm AP, 0.9 mm ML, −6.0 to −7.0 mm DV; for mice—LDT: −5 mm AP, 0.5 mm ML and −3.0 to −4.0 mm DV; NAc: +1.2 mm AP, 0.8 mm ML and −3.2 to −4.2 mm DV. A reference electrode was fixed in the skull, in contact with the dura.

Extracellular neural activity from the LDT and the NAc was recorded using a recording electrode (3–7 MΩ at 1 kHz). Recordings were amplified and filtered by the Neurolog amplifier (NL900D, Digitimer Ltd, UK) (low-pass filter at 500 Hz and high-pass filter at 5 kHz). A recording electrode coupled with a fiber optic patch cable (Thorlabs) was placed in the NAc or LDT. Spontaneous activity of single neurons was recorded to establish baseline for at least 60 s. The DPSS 473 nm laser system (CNI), controlled by a stimulator (Master-8, AMPI), was used for intracranial light delivery and fiber optic output was pre-calibrated to 10–15 mW. Optical stimulation consisted of 80 pulses of 10 ms at 20 Hz. Firing rate was calculated for the baseline, stimulation period and post stimulation period (60 s after the end of stimulation). Neurons showing a firing rate increase or decrease by more than 20% from the mean frequency of the baseline period were considered as responsive, as previously reported, and a heatmap of that response (in percentage) was generated[42]. Additionally, we analyzed the neuronal firing rate distributions in the NAc using the two-sample Kolmogorov-Smirnov test (0.5s bins spanning from 10s before laser stimulation, during laser stimulation of 4 s, through 10s after laser stimulation).

We classified single units in the NAc into three separate groups of putative neurons: putative medium spiny neurons (MSNs), cholinergic interneurons (CINs), and fast-spiking (FS) interneurons, according to previous descriptions[18,19]. FS interneurons were identified has having a waveform half-width of less that 100 µs and a baseline firing rate higher that 10 Hz; tonically active putative CINs (pCINs) were identified as those with a waveform half-width bigger that 300 µs. Putative MSNs (pMSNs) were identified as those with a waveform criterion different from pCIN or pFS and baseline firing rate lower that 5 Hz. Cells that did not fit in any classifications were not considered for the analysis. At the end of each electrophysiological experiment, all brains were collected and processed to identify recording region (Supplementary Fig. 1).

### Behavior

*Two-choice schedule of reinforcement—rats*. During instrumental training, rats are presented two illuminated levers, one on either side of the magazine[24]. Presses on

one lever (Laser+pellet lever (stim+)) leads to instrumental delivery of a pellet plus 4s blue (473 nm) laser stimulation at 10 mW, accompanied by a 4s auditory cue (white noise or tone; always the same paired for a particular rat, but counter-balanced assignments across rats). In contrast, pressing the other lever (pellet alone lever) delivered a single pellet accompanied by another 4s auditory cue (tone or white noise), but with no laser illumination. For both levers, presses during the 4s after pellet delivery have no further consequence. After 2 days of habituation, each daily session begins with a single lever presented alone to allow opportunity to earn its associated reward (either stim+ or pellet Alone), after which the lever is retracted. Then, the alternative lever is presented by itself to allow opportunity to earn the other reward, to ensure that the rat sampled both reward outcomes. Finally, both levers together are extended for the remainder of the session (30 min total), allowing the rat to freely choose between the two levers and to earn respective rewards in any ratio. Whenever the schedule of reinforcements is completed on either lever (FR1, FR4, RR4, RR6), a pellet is immediately delivered, accompanied by 4s of the appropriate auditory cue. For the stim+ lever, delivery of the pellet is also accompanied by additional simultaneous laser stimulation. During those 4s, lever pressing is recorded but no additional stimulation or reward is delivered.

*Two-choice schedule of reinforcement—mice.* Mice were trained for food-seeking operant task and optogenetic stimulation during 20 min daily sessions[43]. Single press on the stim+ lever was paired with the delivery of one 20 mg food pellet (BioServ, USA) and optical stimulations (80 light 10 ms pulses delivered at 20 Hz over 4s) under a FR1. Both ChR2 and eYFP mice received this optical stimulation when food rewards were earned. Single press on the reward alone lever was only paired with the delivery of one 20 mg food pellet (i.e., without optical stimulation) under the same schedule of reinforcement (FR 1). During the acquisition session, cue lights above each lever were on. Responses on the stim+ and the reward alone lever during those 4s periods were recorded but had no additional stimulation or reward. Animals progressed in the behavioral protocol if the acquisition of stable lever-pressing behavior (i.e., <30% variation in lever press activity over three consecutive sessions) was met.

*Progressive ratio.* For both rats and mice, the progressive ratio test was performed with either the stim+ lever or with the pellet alone lever without any laser (order of test conditions is balanced across animals) and repeated for each animal with the other lever[24,43]. The number of presses required to produce the next reward delivery increases after each pressing, according to an exponential progression (progressive ratio schedule: 1, 2, 4, 6, 9, 12, 15, 20, 25, 32, 40, 50, 62, 77, 95, 118, 145, 178, 219, 268, . . .) derived from the formula $PR = [5e^{(reward\ number*0.2)}] - 5$ and rounded to the nearest integer. To determine whether any preference in responding is the result of increased workload, animals are given a FR1 session after PR, identical to the initial day of training.

*Extinction of food—rats.* To conversely assess whether laser stimulation alone can maintain responding on a pellet-laser-associated lever when the reward is discontinued, rats are given the opportunity to earn the same levers but without pellet (pellet extinction)[24]. Each completed trial (RR4) on the stim+ lever results in the delivery of laser stimulation and the previously paired auditory cue but no pellet delivery. Each completed trial on the other lever (previously pellet alone) resulted in the delivery of its auditory cue.

*Extinction of food—mice.* Mice underwent 30 min daily extinction sessions, during which food reward was absent in a FR1 schedule[43]. Each press on the stim+ lever results in the delivery of laser stimulation with no pellet delivery. Presses on the opposite lever (previously pellet alone) had no consequence. Animals were maintained in this condition until behavioral responses were extinguished (<30% variation in lever press activity over three consecutive sessions).

*Extinction of laser stimulation—rats.* To test the persistence of laser-induced preference, rats that have received 2 days reminder training with stim+ and reward versus reward alone[24], underwent 4 consecutive days of laser-extinction testing, where outcomes for both levers consisted in the delivery of a pellet and the associated auditory cue, with no administration of laser stimulation.

*Extinction of laser stimulation—mice.* Following reactivation procedure, animals were tested for 4 days in a persistence procedure (similar as rats)[43], where one lever press on either lever resulted in the delivery of a pellet but with no laser stimulation on the previously associated lever.

*Reversal procedure.* Both rats and mice, after a reminder session on a RR4 (rats) or FR1 schedule (mice), performed a 1-day (30 min) reversal procedure during which the stim+ and reward alone levers were switched. In this paradigm, a single response on the previously inactive lever (reward alone lever) was paired to the delivery of phasic optical stimulations.

*Conditioned place preference.* The CPP protocol was adapted from a previously published report[3]. The CPP apparatus consisted of two compartments with different patterns, separated by a neutral area (Med Associates): a left chamber measuring 27.5 cm × 21 cm with black walls and a grid metal floor; a center chamber measuring 15.5 cm × 21 cm with gray walls and gray plastic floor; and a right chamber measuring 27.5 cm × 21 cm with white walls and a mesh metal floor. Rat location within the apparatus during each preference test was monitored using a computerized photo-beam system (Med Associates). Briefly, on day 1, individual rats were placed in the center chamber and allowed to freely explore the entire apparatus for 15 min (pre-test). On day 2, rats were confined to one of the side chambers for 30 min and paired with optical stimulation, ON side; in the second session, rats were confined to the other side chamber for 30 min with no stimulation, OFF side. Conditioning sessions were counterbalanced. On day 3 rats were allowed to freely explore the entire apparatus for 15 min (post-test). Optical stimulation consisted of 80 pulses of 10 ms at 20 Hz, every 15 s with a blue laser and 4 s of constant light at 10 mW with a yellow laser. Results are expressed as the total time, ratio of preference and the difference of time spent in each side.

*Real-time place preference.* RTPP test was performed in a custom-made black plastic arena (60 × 60 × 40 cm) comprised by two indistinguishable chambers, for 15 min One chamber was paired with light stimulation of 10 ms pulses at 20 Hz for the excitation experiments (Chr2) and constant yellow light at 10 mW for inhibition experiments (NpHR) during the entire period that the animal stayed in the stimulus-paired side. The choice of paired chamber was counterbalanced across rats. Animals were placed in the no-stimulation chamber at the start of the session and light stimulation started at every entry into the paired chamber. Animal activity was recorded using a video camera and time spent in each chamber was manually assessed. Results are presented as percentage of time spent in each chamber and the difference of time spent in each side.

*Locomotor activity.* Animals attached to an optical fiber connected to a laser were placed in the center of an arena (43.2 cm × 43.2 cm; Med Associates). Locomotion was monitored online over a period of 30 min Stimulation consisted of 80 light pulses of 10 ms (473 nm; blue laser) delivered at 20 Hz or 4 s of constant yellow light every 15 s. Total distance traveled was used as indicator of locomotor activity.

*Food consumption.* Food consumption test was conducted in a familiar chamber containing bedding; rats had serial access to pre-weighed quantities of regular chow pellets (20–22 g; 4RF21, Mucedola SRL) and palatable food pellets (20–22 g; F0021, BioServ) while also having constant access to water. Each food intake session consisted of 20 min access to 20 g of regular chow followed by 20 min of access to 20–22 g of palatable food pellets and chow. Laser conditions were the same as above. Food consumption was repeated on 3 consecutive days. Laser stimulation was administered only on of the days (either day 2 or 3; counterbalanced across rats). Control intake was measured in the absence of any laser stimulation on the remaining days. Chow and palatable food pellets were reweighed at the end of the test to calculate total consumption.

*Immunofluorescence.* For c-fos analysis, animals were anaesthetized with pentobarbital (Eutasil, Lisbon, Portugal) 90 min after initiation of the PR test, and transcardially perfused with 0.9% saline followed by 4% paraformaldehyde. Brains were removed and sectioned coronally at a thickness of 50 μm, on a vibrating microtome (VT1000S, Leica, Germany).

For the quantification of different LDT-NAc projections, animals were sacrificed similarly but without performing any behavioral test.

Sections were incubated with the primary antibody mouse anti-D2 receptor (1:500, B-10, Santa Cruz Biotechnology); rabbit anti-c-fos (1:1,000, Ab-5, Merck Millipore), goat anti-GFP (1:500, ab6673, Abcam), mouse anti-D1 receptor (1:100, NB110–60017, Novus), rabbit anti-GAD65+GAD67 (1:1000, ab11070, Abcam), mouse anti-EAAC1 (1:500, MAB1587, Millipore), goat anti-ChAT (1:750, AB144P, Millipore) or sheep anti-ChAT (1:500, ab18736, Abcam) followed by appropriate secondary fluorescent antibodies (Alexafluor, 1:1000, Invitrogen, MA, USA). All sections were stained with 4′,6-diamidino-2-phenylindole (DAPI; 1 mg ml$^{-1}$) and mounted using mounting media (Permafluor, Invitrogen, MA, USA).

Positive cells within the brain regions of interest were analyzed and cell counts were performed by confocal microscopy (Olympus FluoViewTMFV1000). Estimation of cell density was obtained by dividing cell number values with the corresponding areas, determined using an Olympus BX51 optical microscope and the StereoInvestigator software (Microbrightfield).

**Statistical analysis.** Statistical analysis was performed in GraphPad Prism 5.0 (GraphPad Software, Inc., La Jolla, CA, USA) and SPSS Statistics v19.0 (IBM corp., USA). Parametric tests were used whenever Shapiro-Wilk normality test SW>0.05. Repeated measures one way or two-way analysis of variance (ANOVA) was used when appropriate. Bonferroni's post hoc multiple comparison tests were used for group differences determination. Statistical analysis between two groups was made using Student's *t*-test. We then compared (baseline versus stimulation) the neuronal firing rate distributions in the NAc for all neurons using the two-sample Kolmogorov-Smirnov test (0.5s bins spanning from 10s before laser stimulation, during laser stimulation of 4s, through 10s after laser stimulation).

Pearson's correlation was used to examine the relationship between recruited c-fos+ cells and breakpoint levels reached in the progressive ratio task. Results are presented as mean ± s.e.m. Statistical significance was accepted for $p < 0.05$.

**Reporting summary**. Further information on research design is available in the Nature Research Reporting Summary linked to this article.

## Data availability
The data that support the findings of this study are available from the corresponding author upon reasonable request.

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

## Acknowledgements
AJR was a FCT investigator (IF/00883/2013). B.C., C.S.-C., and S.B. are recipients of Fundação para a Ciência e Tecnologia (FCT) fellowships (SFRH/BD/51992/2012; SFRH/BD/98675/2013; SFRH/BD/90374/2012; SFRH/BD/89936/2012). This work was developed under a BIAL Foundation Grant (PT/FB/BL-2016-030). Part of the work was supported by FCT project PTDC/MED-NEU/29071/2017. This project is in the scope of the project NORTE-01-0145-FEDER-000013, supported by the Northern Portugal Regional Operational Programme (NORTE 2020), under the Portugal 2020 Partnership Agreement, through the European Regional Development Fund (FEDER). Part of this work has also been funded by FEDER funds, through the Competitiveness Factors Operational Programme (COMPETE), and by National funds, through the Foundation for Science and Technology (FCT), under the scope of the project POCI-01-0145-FEDER-007038.

## Author contributions

A.J.R., N.S., and B.C. designed experiments and wrote the manuscript; B.C. performed, and B.C. and N.A.P.V. analyzed electrophysiological data; B.C. performed optogenetic experiments and B.C., C.S.C., S.B., and A.V.D. performed behavioral evaluation; B.C. performed the immunofluorescence analysis. A.J.R. and N.S. provided funding and infrastructures. All authors read and approved the final version of the manuscript.

## Additional information

**Competing interests:** The authors declare no competing interests.

