## [Peer Review File · Nature Communications]

Reviewers' Comments:

Reviewer #1:

Remarks to the Author:

The study by Coimbra and colleagues present a series of interesting and novel findings, that identify the projections between laterodorsal tegmentum and nucleus accumbens as a major driver that is both necessary and sufficient for motivation and preference.

The authors show in elegant and well-controlled experiments that:

- Projections between the laterodorsal tegmentum and nucleus accumbens are predominantly excitatory
- Optogenetic activation of this projection is sufficient to drive motivation, induce place preference, and induce self-stimulation.
- Optogenetic inhibition of this projection reveals that this projection is necessary to drive motivation and place preference.
- The cholinergic projections between the laterodorsal tegmentum and nucleus accumbens are predominantly excitatory and their stimulation recapitulates the increases in motivation and place preference observed.

The studies are well done, and the conclusions clear. The main puzzlement one is left with is the following" Other projection pathways to the nucleus accumbens have been reported to do have a similar effect. So one should expect that any signal exciting ccumbens in a reward-related way could produce similar effects? or is this projection unique? Maybe the authors could comment on this?

Reviewer #2:

Remarks to the Author:

The LDT has been implicated in reward related behaviors. There have been a few papers that have looked at which specific LDT projections might mediate their effect, and most have focused on the VTA. Given that the LDT also projects to the NAc, the authors set out to determine how LDT projections to the NAc influence reward related behaviors. The major strength of the paper is a systematic exploration of a large number of behavioral assays across two species and three different transgenic lines: ChAT, vglut2, and vgat. In general the effects of nonspecific LDT-NAc stimulation are complementary to LDT-VTA effects and were most replicated by ChAT neuron specific LDT-NAc stimulation. I think these experiments are a valuable contribution to the literature.

Major comments:

The controls used in the rat studies are unconventional. From my understanding, the authors used only only cre-dependent ChR2 in the LDT as the control group rather than a fluorophore. This would mean that experimental animals received an additional WGA-Cre injection. Additionally there appears to be only 1 control group for the two different experimental groups: ChR2 and eNpHR. These experimental groups had different stimulation parameters: ChR2 pulsed blue light, eNpHR constant yellow light. Which stimulation did the control group receive? More information about the controls should be disclosed. If there are confounds present they should be acknowledged.

Stats: The authors note that they used a 2-way ANOVA for many of their tests. I am not sure whether they used a repeated measures ANOVA to account for the repeated measurements from individual subjects. A longitudinal analytic technique, either repeated measures ANOVA or linear mixed effects model should be used. I suspect the main results would remain significant.

I would appreciate it if the authors included more information about the time course of optogenetic stimulation effects on NAc firing rates, given that acetylcholine may have both short and longer term effects. I hope that this would not be a significant burden on the authors. One possibility would be a "heatmap" of firing rates for all neurons in <1 sec bins for >10 sec before stim, 4 sec during stim, and >10 sec post stim, to give a sense of the peak and offset dynamics.

Methods: I could not find information about how some of the tasks were performed. In particular, I think it would be helpful to provide more detail on the equipment used for RTPP and to describe how food consumption was tested.

Minor comments:

In a paper with a number of interesting results, one that I particularly find fascinating is that LDT-NAc stim is sufficient to drive ICSS, but much more potently modifies lever pressing for a reward stimulus, even in laser extinction. I appreciate the authors' discussion about this finding. I wish that there had been further exploration of this and hope that it will be explored in future studies.

Given that one of the strengths of the paper is the systematic and repeated nature of the experiments, it would be helpful if the figures also followed the same conventions between different groups. In general the authors do this, but there are a few differences between main figures and supplemental figures, and some unclear choices regarding differences in axes between groups, e.g. SupFig5b, SupFig6b.

How well does AAV5-EF1a-WGA-Cre-mCherry work as a specific retrograde virus? I am concerned that it may have transsynaptic effects and wonder when it may work best. It is not clear why 4 weeks was chosen as the post-injection time frame. Ultimately these results are supported in other ways including prior work with CTB.

SupFig2: B&C: It is not clear why the solid/dashed lines for the control mice are reversed between the two panels. It is also not clear what comparison is being tested by the asterisks *.

Why were mice trained only using an FR1 schedule whereas rats had mixed FR & RR schedules?

I find the results of Fig3D surprising and interesting. Why would presses on the non-laser lever be so high for the NpHR group?

Given that the major strength of the paper was the systematic nature of the behaviors tested, I can understand why behaviors were done in the same order across animals. It perhaps would have been ideal to counterbalance order to ensure that one did not influence the next.

Reviewer #3:

Remarks to the Author:

Coimbra et al demonstrate a functional role for the recently-identified cholinergic projection from the laterodorsal tegmental nucleus (LDT) to the nucleus accumbens (NAc). Using retrogradely-transported cre (delivered via WGA-AAV injected in the NAc) and cre-dependent expression of channelrhodopsin or halorhodopsin via AAV injection in the LDT, they show that stimulation or inhibition of LDT terminals in the NAc influences performance of several tasks. This work is extensive in that several behavioral tests were used, as well as manipulations to target the LDT-NAc projection in several ways (non-

specifically, and the specific cholinergic, glutamatergic and GABAergic projections). The study nicely demonstrates potential functional roles of the cholinergic novel projection, whereas stimulation of the glutamatergic and GABAergic projections has less pronounced effects. However, despite the significance of the findings, there are a number of odd results that need further explanation, and there are some data analysis and interpretation problems that should be corrected, as described below.

1. The large virus injections (up to 1 microliter) likely spread out of the LDT to adjacent brain areas (e.g., the periaqueductal grey). If neurons in these areas project to the NAc, some of the terminals stimulated in the NAc could come from non-LDT neurons. This is of particular concern because the significance of the paper is due to the novelty of the LDT-NAc projection. Please provide histological evidence to address whether this could have been the case. (The micrographs shown do not include adjacent brain regions, and a numerical analysis of positive cells in LDT vs adjacent structures is not provided.) Furthermore, it's even possible that some of the virus entered the ventricle, in which case the terminals stimulated in the NAc could come from a large number of different possible regions. Placement controls (e.g., inject cre-dependent Chr2 AAV in the PAG or ventricle, and WGA-Cre in NAc) would alleviate these concerns.

2. Behavior in the two-lever choice task includes some components that need further explanation and analysis. As shown in Fig. 1b and 2b, without active Chr2 or Halo, the maximum number of lever presses on RR6 is a bit over 400 (stim- lever plus stim+ lever). However, in rats with Chr2 or halo, the maximum seems to be greater than this. My observation here could be incorrect due to the inaccuracy of adding numbers estimated from the graph. However, the apparent oddity here underscores that it would be useful to see some additional measures of performance: fraction of lever presses on the stim+ and stim- levers, and total number of lever presses. If the total number of lever presses is really greater in Chr2 and/or Halo, this raises the question of why that is. Do animals press the lever faster? Or resume pressing sooner after a reward? And are these vigor measures different for the different levers, or different after the animals receive stimulation vs pellet alone?

3. Fig. 1d shows that, during laser extinction, responding on the lever previously associated with the laser+pellet grows from <600 presses per session to >800, over the course of four RR4 sessions. Why? Clearly the rate of responding is increasing, as the number of lever presses on the stim- lever remains constant. What aspect of vigor (see #2) is increasing? And what might be the biological/mechanistic reason for the increase? Quite remarkably, there is a similar increase in fig. 2d for the previously halo-associated lever – please comment on this.

4. The ICSS results do not inspire confidence. First, at most 50 lever presses per hour (<1 press per min) is quite weak. This was the strongest effect, and it is barely greater than ICSS in the control condition. Second, it's possible that different stimulation parameters would produce stronger ICSS. Finally, it seems that the light stimulation protocol was different for ICSS vs the other tasks, so it's hard to compare this result to, say, the pellet extinction part of the choice task. Perhaps these results should be removed, or moved to supplemental.

5. For progressive ratio tasks, the break point apparently is defined as the maximum number of lever presses. However, for statistical (and graphical display) purposes, the maximum number of rewards earned should be used because the number of lever presses is, by definition, not normally distributed (as the function escalates exponentially). Alternatively, the authors could use non-parametric statistics.

6. Some bar graphs (e.g., in fig. 2) display individual data points. The remaining bar graphs (e.g., fig. 1) should also.

7. For the extinction test, the authors refer to "the first trial." But I think "the first session" is meant.
8. The timeline in fig S1 should include surgery, recovery, waiting for virus expression, etc.
9. in fig. S3, what is the relevance/meaning of the spike latency to optical inhibition? And why is it so short?
10. Fig s4 needs a color legend.
11. Fig s7 should be better integrated into the main paper (i.e., the results should be described fully and not just mentioned in the Discussion), and perhaps not buried in supplemental.

Response to reviewers

Authors would like to thank the reviewers for the pertinent and constructive comments which helped to clarify and improve this manuscript. We have performed additional experiments and included reviewers' suggestions in this new version; we believe we have addressed all the remarks.

Reviewer #1 (Remarks to the Author):

The study by Coimbra and colleagues present a series of interesting and novel findings, that identify the projections between laterodorsal tegmentum and nucleus accumbens as a major driver that is both necessary and sufficient for motivation and preference.

The authors show in elegant and well-controlled experiments that:

- Projections between the laterodorsal tegmentum and nucleus accumbens are predominantly excitatory
- Optogenetic activation of this projection is sufficient to drive motivation, induce place preference, and induce self-stimulation.
- Optogenetic inhibition of this projection reveals that this projection is necessary to drive motivation and place preference.
- The cholinergic projections between the laterodorsal tegmentum and nucleus accumbens are predominantly excitatory and their stimulation recapitulates the increases in motivation and place preference observed.

The studies are well done, and the conclusions clear. The main puzzlement one is left with is the following " Other projection pathways to the nucleus accumbens have been reported to do have a similar effect. So, one should expect that any signal exciting accumbens in a reward-related way could produce similar effects? or is this projection unique? Maybe the authors could comment on this?

Authors: The reviewer raised a very pertinent point regarding the functional role of this specific projection versus other excitatory projections to the nucleus accumbens (NAc). Previous experiments¹ showed that not all excitatory inputs to NAc originate the same behavioural outcome. Similarly, herein, we show that glutamatergic LDT-NAc inputs induce a net excitatory response in NAc neurons and increased preference for the lever associated with stimulation (Fig. 5b, 5f), but to a less extent than the LDT-NAc cholinergic inputs stimulation. In addition, we even observe an aversive effect of the stimulus on the RTPP paradigm (Fig. 5k). So, we believe that even though there are different excitatory signals arriving to the NAc, their intensity and their behavioral effect depends on the type of the input. Accordingly, and as the reviewer suggested, we now discussed this topic in the revised version of the manuscript (page 11).

Reviewer #2 (Remarks to the Author):

The LDT has been implicated in reward related behaviors. There have been a few papers that have looked at which specific LDT projections might mediate their effect, and most have focused on the VTA. Given that the LDT also projects to the NAc, the authors set out to determine how LDT projections to the NAc influence reward related behaviors. The major strength of the paper is a systematic exploration of a large number of behavioral assays across two species and three different transgenic lines: ChAT, vglut2, and vgat. In general the effects of nonspecific LDT-NAc stimulation are complementary to LDT-VTA effects and were most replicated by ChAT neuron specific LDT-NAc stimulation. I think these experiments are a valuable contribution to the literature.

Major comments:

The controls used in the rat studies are unconventional. From my understanding, the authors used only cre-dependent ChR2 in the LDT as the control group rather than a fluorophore. This would mean that experimental animals received an additional WGA-Cre injection. Additionally there appears to be only 1 control group for the two different experimental groups: ChR2 and eNpHR. These experimental groups had different stimulation parameters: ChR2 pulsed blue light, eNpHR constant yellow light. Which stimulation did the control group receive? More information about the controls should be disclosed. If there are confounds present they should be acknowledged.

Authors: We thank the reviewer for this comment, since the previous description of the methods was not clear. We evaluated 3 sets of rats: in the first set, we also included a group with only cre-dependent ChR2 in the LDT, but in the subsequent two sets, control groups were always injected with cre-dependent YFP in the LDT + WGA-cre in the NAc. Because we had no effect of stimulation in any of the control YFP groups, we joined the animals for presentation purposes in the previous version of the manuscript.

In this revised version of the manuscript, we have only included data from the two sets of animals with the YFP group. So, YFP animals of ChR2 experiments were stimulated with blue laser, and those of NpHR inhibition experiments were stimulated with yellow laser. This is now clearly explained in the methods section. All rat behavioral data is different from the previous version but the results are the same (Fig. 2-3).

Stats: The authors note that they used a 2-way ANOVA for many of their tests. I am not sure whether they used a repeated measures ANOVA to account for the repeated measurements from individual subjects. A longitudinal analytic technique, either repeated measures ANOVA or linear mixed effects model should be used. I suspect the main results would remain significant.

Authors: We agree and revised the manuscript accordingly. We have now clearly referred when repeated measures ANOVA was performed.

I would appreciate it if the authors included more information about the time course of optogenetic stimulation effects on NAc firing rates, given that acetylcholine may have both short and longer term effects. I hope that this would not be a significant burden on the authors. One possibility would be a "heatmap" of firing rates for all neurons in <1 sec bins for >10 sec before stim, 4 sec during stim, and >10 sec post stim, to give a sense of the peak and offset dynamics.

Authors: As suggested, we have analysed the firing rates of all neurons in 0.5s bins, >10 sec before stim, 4 sec during stim, and >10 sec post stim, and created heatmaps for all figures with electrophysiological data. We believe that now the effects of stimulation/inhibition are clearer to the reader (Fig. 1k, Fig. 4d, Fig. 5d, Fig. 6d).

Methods: I could not find information about how some of the tasks were performed. In particular, I think it would be helpful to provide more detail on the equipment used for RTPP and to describe how food consumption was tested.

Authors: We are sorry for this omission. We have now added information about food consumption and locomotor activity in the methods. The RTPP arena was custom made, as described in the methods.

Minor comments:

In a paper with a number of interesting results, one that I particularly fascinating is that LDT-NAc stim is sufficient to drive ICSS, but much more potently modifies lever pressing for a reward stimulus, even in laser extinction. I appreciate the authors' discussion about this finding. I wish that there had been further exploration of this and hope that it will be explored in future studies.

Authors: We also think that these findings are interesting, and we are planning experiments to address this in the future.

Given that one of the strengths of the paper is the systematic and repeated nature of the experiments, it would be helpful if the figures also followed the same conventions between different groups. In general the authors do this, but there are a few differences between main figures and supplemental figures, and some unclear choices regarding differences in axes between groups, e.g. SupFig5b, SupFig6b.

Authors: Thank you for pointing this out. We have now changed the figures, and included data in a systematic manner, please see new figures.

How well does AAV5-EF1a-WGA-Cre-mCherry work as a specific retrograde virus? I am concerned that it may have transsynaptic effects and wonder when it may work best. It is not clear why 4 weeks was chosen as the post-injection time frame. Ultimately these results are supported in other ways including prior work with CTB.

Authors: The original manuscript that described the WGA-cre construct evaluated animals 5 week post-injection². However, in a previous work focusing on LDT-striatum, researchers shortened to 4 weeks: "The time between injections and perfusion was shortened compared with previous studies (see Materials and Methods; Gradinaru et al., 2010; Xu and Südhof, 2013) to minimize the possibility of transsynaptic retrograde labelling of second-order neurons at the level of the thalamus and the VTA. However, because we cannot rule out this possibility entirely, we performed additional experiments using double retrograde tracer injections into the NA core and either the VTA ($n = 2$) or the mediodorsal thalamus ($n = 2$)"³.

Our tracing data showing that around 50% of LDT-NAc neurons are cholinergic (Fig. 1e), is supported by this previous work which shows 59-74% ChAT+ projections³. In this new version, we included this information (page 3):

"Nearly 50% of YFP-transfected LDT neurons were cholinergic, 29% glutamatergic and 23% GABAergic (Fig. 1d-e); in line with previous observations showing that 59-74% of LDT-NAc projections were cholinergic³."

SupFig2: B&C: It is not clear why the solid/dashed lines for the control mice are reversed between the two panels. It is also not clear what comparison is being tested by the asterisks*.

Authors: We are sorry for the mistake, that we have now corrected - In this new version the values are slightly different because we included the two set of animals in rat behavioral tests.

Why were mice trained only using an FR1 schedule whereas rats had mixed FR & RR schedules?

Authors: These behavioral tests were based on previous published papers, which had some differences in the type of training, depending on the species used. Rats – protocol described by Robinson *et al.*⁶; Mice – protocol described by Adamantidis *et al.*⁷

I find the results of Fig3D surprising and interesting. Why would presses on the non-laser lever be so high for the NpHR group?

Authors: We thank the reviewer for bringing attention to this, which we also find very interesting. One possible explanation is that animals were previously exposed to pellet extinction conditions, so in these sessions in which the reward is present again, they readapt their behavior and increase the number of lever presses in the previously preferred lever. We now discuss this in more detail in page 11. (please see also response to reviewer 3 comment below).

Given that the major strength of the paper was the systematic nature of the behaviors tested, I can understand why behaviors were done in the same order across animals. It perhaps would have been ideal to counterbalance order to ensure that one did not influence the next.

Authors: The instrumental behaviors have to be in this specific order due to the nature of the different tests. Yet, we could have performed the CPP and RTPP first and after the instrumental behavior, but unfortunately, we did not change the order of the tests. Still, we do not believe that the order has a dramatic effect in these particular behaviours.

Reviewer #3 (Remarks to the Author):

Coimbra et al demonstrate a functional role for the recently-identified cholinergic projection from the laterodorsal tegmental nucleus (LDT) to the nucleus accumbens (NAc). Using retrogradely-transported cre (delivered via WGA-AAV injected in the NAc) and cre-dependent expression of channelrhodopsin or halorhodopsin via AAV injection in the LDT, they show that stimulation or inhibition of LDT terminals in the NAc influences performance of several tasks. This work is extensive in that several behavioral tests were used, as well as manipulations to target the LDT-NAc projection in several ways (non-specifically, and the specific cholinergic, glutamatergic and GABAergic projections). The study nicely demonstrates potential functional roles of the cholinergic novel projection, whereas stimulation of the glutamatergic and GABAergic projections has less pronounced effects. However, despite the significance of the findings, there are a number of odd results that need further explanation, and there are some data analysis and interpretation problems that should be corrected, as described below.

1. The large virus injections (up to 1 microliter) likely spread out of the LDT to adjacent brain areas (e.g., the periaqueductal grey). If neurons in these areas project to the NAc, some of the terminals stimulated in the NAc could come from non-LDT neurons. This is of particular concern because the significance of the paper is due to the novelty of the LDT-NAc projection. Please provide histological evidence to address whether this could have been the case. (The micrographs shown do not include adjacent brain regions, and a numerical analysis of positive cells in LDT vs adjacent structures is not provided.) Furthermore, it's even possible that some of the virus entered the ventricle, in which case the terminals stimulated in the NAc could come from a large number of different possible regions. Placement controls (e.g., inject cre-dependent ChR2 AAV in the PAG or ventricle, and WGA-Cre in NAc) would alleviate these concerns.

Authors: This is a pertinent point that we believe we have addressed in this version of the manuscript:

1. First, we provide histological images of 3 representative animals from all the experimental groups, showing that there is minor viral spreading to LDT adjacent areas.

In Sup. Fig 2. and Sup. Fig. 3, we have now added representative figures from different animals of each group (rats, mice), and from 3 different positions to the bregma, to show the

viral expression in the LDT (and adjacent areas). The expression is mostly restricted to the LDT, though some YFP+ cells can be seen in the PAG. One animal presented very few YFP+ cells in the dorsal tegmentum nucleus (DT).

2. Second, and as suggested by the reviewer, we created two additional control groups to verify the specificity of our approach:
 - a. Group 1: animals injected with WGA-cre in NAc + DIO-ChR2 in the PAG (coordinates from bregma: -8.2 mm AP, 0.9 mm ML, -5 mm DV)
 - b. Group 2: animals injected with WGA-cre in NAc + DIO-ChR2 in the 4th ventricle (coordinates from bregma: -8.52 mm AP, 0.0 mm ML, -5.5 mm DV).

Group 1 results: Histological evaluation of PAG-injected animals revealed no YFP+ cell bodies in the PAG nor YFP+ terminals in the NAc (Figure 1 below). These results confirm the literature, since PAG-NAc direct connections have not been described to date. Still, we stimulated “hypothetical” PAG terminals in the NAc (using the same coordinates as for other experiments); and did not observe any behavioral effect whatsoever (Figure 2 below).

Group 2 results: Histological evaluation of 4V-injected animals revealed no labelling in any of the 4V adjacent areas (Figure 1 below). Sparse labelling of the cerebellum was found in one animal. In all of the tested animals, we found no YFP+ terminals in the NAc. Even in the absence of YFP terminals in the histological analysis, we also stimulated “hypothetical” terminals and did not observe any behavioral effect (Figure 2 below).

2. Behavior in the two-lever choice task includes some components that need further explanation and analysis. As shown in Fig. 1b and 2b, without active ChR2 or Halo, the maximum number of lever presses on RR6 is a bit over 400 (stim- lever plus stim+ lever). However, in rats with ChR2 or halo, the maximum seems to be greater than this. My observation here could be incorrect due to the inaccuracy of adding numbers estimated from the graph. However, the apparent oddity here underscores that it would be useful to see some additional measures of performance: fraction of lever presses on the stim+ and stim- levers, and total number of lever presses. If the total number of lever presses is really greater in ChR2 and/or Halo, this raises the question of why that is. Do animals press the lever faster? Or resume pressing sooner after a reward? And are these vigor measures different for the different levers, or different after the animals receive stimulation vs pellet alone?

Authors: Please see response to your comment 3 below.

3. Fig. 1d shows that, during laser extinction, responding on the lever previously associated with the laser+pellet grows from <600 presses per session to >800, over the course of four RR4 sessions. Why? Clearly the rate of responding is increasing, as the number of lever presses on the stim- lever remains constant. What aspect of vigor (see #2) is increasing? And what might be the biological/mechanistic reason for the increase? Quite remarkably, there is a similar increase in fig. 2d for the previously halo-associated lever – please comment on this.

Authors: We completely agree with the reviewer that the dynamics of response is a very relevant point that needs further exploration. But unfortunately, due to technical limitations, our set up only allowed to register the total number of lever presses per session, and did not register lever presses/time. This an obvious limitation of the data, which we now discuss in page 11.

Regarding the total number of lever presses, this data is now shown in Sup. Fig 5a-b, in which the reviewer can observe that there are no differences in total number of lever presses between YFP and ChR2 or NpHR groups. Fraction of lever presses is also provided (Sup. Fig. 5 c-d).

Considering the last point raised by the reviewer, we also found very curious the observed increase in lever presses in stim+ in laser extinction conditions for ChR2 group (or in stim-lever for NpHR group). As stated in the previous response, we did not register the pattern of lever press, which compromises our ability to speculate on the biological/mechanistic reason for the increase. Still, we discuss a possible hypothesis in this new version of the manuscript (page 11):

“Interestingly, in laser extinction conditions (two levers yield similar reward), ChR2 animals still present a robust preference for the stim+ lever though no stimulation is given. Of notice, ChR2 animals even slightly increase the number of presses in this lever (conversely, NpHR animals increase presses in stim- lever). One can hypothesize that since these animals were previously exposed to pellet extinction conditions, now, when the reward is present again, they readjust the pattern of lever presses in the previously preferred (reward-associated) lever. In other words, it can reflect enhanced salience of the preferred lever when the reward is available again. Unfortunately, in this present study we did not register the pattern of lever press, which would be important to better understand these findings.”

4. The ICSS results do not inspire confidence. First, at most 50 lever presses per hour (<1 press per min) is quite weak. This was the strongest effect, and it is barely greater than ICSS in the control condition. Second, it's possible that different stimulation parameters would produce stronger ICSS. Finally, it seems that the light stimulation protocol was different for ICSS vs the other tasks, so it's hard to compare this result to, say, the pellet extinction part of the choice task. Perhaps these results should be removed, or moved to supplemental.

Authors: As suggested, we removed the ICSS data from this manuscript. We believe that the other tests already evidence the role of LDT-NAc projections in rewarding behavior.

5. For progressive ratio tasks, the break point apparently is defined as the maximum number of lever presses. However, for statistical (and graphical display) purposes, the maximum number of rewards earned should be used because the number of lever presses is, by definition, not normally distributed (as the function escalates exponentially). Alternatively, the authors could use non-parametric statistics.

Authors: We understand the reviewer's point, and have now also included the number of rewards earned in all progressive ratio (PR) tests, in addition to the breakpoint.

Yet, we would like to clarify that the breakpoint is not the maximum number of lever presses of the entire session, rather is the maximum number of lever presses that the animals performed in the last trial of the session. For example, most of the animals reach trial n°13 (meaning they received 13 food pellets), which implicates that they had to press the lever 62 times to reach this point (number of cumulative presses = 216). The next trial, in order to receive the 14th food pellet, animals would have to press 77 additional lever presses, which is already a significant effort for a single pellet (Table 1). So, one can have two animals which received 14 pellets but have different breakpoint values.

Regarding the last part of the comment, since we only use the number of lever presses in the last trial of the session, and data passes normality distribution test (Shapiro-Wilk test of normality), we used parametric statistics (Repeated-measures 2way ANOVA).

Table 1. Example of a PR task in rodents.

Level	Level trial 12	Level trial 13	Level trial 14	Level trial 15	Level trial 16
Cumulative presses Within the level	216	278	355	450	568

Presses to go to next level	62	77	95	118	145
Pellets earned	12	13	14	15	16

6. Some bar graphs (e.g., in fig. 2) display individual data points. The remaining bar graphs (e.g., fig. 1) should also.

Authors: Yes, we agree. Please see this new version.

7. For the extinction test, the authors refer to “the first trial.” But I think “the first session” is meant.

Authors: We agree. Revised.

8. The timeline in fig S1 should include surgery, recovery, waiting for virus expression, etc.

Authors: Agree. Revised. Please see this new version (Supplementary Fig. 1a).

9. in fig. S3, what is the relevance/meaning of the spike latency to optical inhibition? And why is it so short?

Authors: This latency refers to the neurons that still fire (do not respond) to inhibition. But since this can be misleading, we removed this graph from the manuscript.

10. Fig s4 needs a color legend.

Authors: Revised.

11. Fig s7 should be better integrated into the main paper (i.e., the results should be described fully and not just mentioned in the Discussion), and perhaps not buried in supplemental.

Authors: Our decision was based on the few data we had regarding D1- and D2-MSNs, which we felt was insufficient to drive many conclusions about this.

But we also agree that this is an interesting finding, and thus have now included the description of this data in the results section (page 6-7) and discussed it in more detailed in this new version of the manuscript, as suggested by the reviewer.

Figure 1. Histological evaluation of control groups. a) One group of animals was injected with WGA-cre in NAc + DIO-ChR2 in the PAG. b) No YFP expression was found in the PAG nor in the NAc. b) One other group of animals was injected with WGA-cre in NAc + DIO-ChR2 in the 4V. d) No expression of YFP+ cells in the 4V nor adjacent areas, nor YFP+ terminals in the NAc. Scale bar = 0.5mm

Figure 2. Behavioral evaluation of control groups. a) Strategy used in order to stimulate PAG hypothetical terminals in the NAc. One group of animals was injected with WGA-cre in NAc + DIO-ChR2 in the PAG. b) No expression of YFP+ cells in the PAG, nor YFP+ terminals in the NAc (c), as expected. Scale bar = 1mm. d) Behavioral evaluation of control PAG group. Stimulation of hypothetic PAG terminals in the NAc did not induce any preference in the two choice task. e) All groups decrease responding in pellet extinction conditions. f,g) No effect of PAG-NAc optogenetic activation in the progressive ratio test ($n_{\text{PAG-ChR2}}=5$, $n_{\text{PAG-YFP}}=4$). h) Strategy used to control for misplaced injection in the 4V. One group of animals was injected with WGA-cre in NAc + DIO-ChR2 in the 4V. i) No expression of YFP+ cells in the 4V, nor YFP+ terminals in the NAc (j). Scale bar = 1mm. k) Behavioral evaluation of control 4V group. Stimulation of hypothetic terminals in the NAc did not induce any preference in the two choice task. l) All groups decrease responding in pellet extinction conditions. m,n) No effect of optogenetic activation in the progressive ratio test ($n_{4\text{V-ChR2}}=6$, $n_{4\text{V-YFP}}=5$). Data represented as mean \pm SEM.

Bibliography

1. Stuber, G. D. *et al.* Excitatory transmission from the amygdala to nucleus accumbens facilitates reward seeking. *Nature* **475**, 377–380 (2011).
2. Gradinaru, V. *et al.* Molecular and Cellular Approaches for Diversifying and Extending Optogenetics. *Cell* **141**, 154–165 (2010).
3. Gradinaru, V. *et al.* Molecular and Cellular Approaches for Diversifying and Extending Optogenetics. *Cell* **141**, 154–165 (2010).
4. Xu, W. & Südhof, T. C. A neural circuit for memory specificity and generalization. *Science* **339**, 1290–1295 (2013).
5. Dautan, D. *et al.* A major external source of cholinergic innervation of the striatum and nucleus accumbens originates in the brainstem. *J. Neurosci. Off. J. Soc. Neurosci.* **34**, 4509–4518 (2014).
6. Robinson, M. J. F., Warlow, S. M. & Berridge, K. C. Optogenetic Excitation of Central Amygdala Amplifies and Narrows Incentive Motivation to Pursue One Reward Above Another. *J. Neurosci.* **34**, 16567–16580 (2014).
7. Adamantidis, A. R. *et al.* Optogenetic interrogation of dopaminergic modulation of the multiple phases of reward-seeking behavior. *J. Neurosci. Off. J. Soc. Neurosci.* **31**, 10829–10835 (2011).

Reviewers' Comments:

Reviewer #1:

Remarks to the Author:

The authors have addressed my concerns in a satisfactory manner.

Reviewer #2:

Remarks to the Author:

I appreciate the opportunity to read the revision of this manuscript. The authors have tried to address all of my prior questions, and I commend their work. I only have one significant point of clarification remaining regarding the new heatmaps. On most there is a striking similarity between neurons. For example Fig1k (top 2 rows appear identical, nearly identical for next 2 rows, another 2 sets of rows appear identical around ~50, and again around ~38, etc), Fig4d (alternating rows on the top have striking similarity during stimulation, rows around 10 are very similar during baseline), Fig5d (many pairs of lines have striking similarities, esp pre and post), Fig6d (strong similarities both on top and bottom). n's would be helpful in contextualizing this: how many trials from how many neurons from how many animals, and how many neurons were recorded simultaneously. I could only easily find the total number of neurons in each experiment. But even with more information about n's, there needs to be some other explanation of why so many lines are so similar.

Reviewer #3:

Remarks to the Author:

The authors have addressed my concerns. They may wish to consider including the graphs in their rebuttal in the supplementary data as these results strengthen the paper.

Response to reviewers

Authors would like to thank the reviewers for the constructive comments and for the fair evaluation of the manuscript. Please find below our remarks.

Reviewer #2 (Remarks to the Author):

I appreciate the opportunity to read the revision of this manuscript. The authors have tried to address all of my prior questions, and I commend their work. I only have one significant point of clarification remaining regarding the new heatmaps. On most there is a striking similarity between neurons. For example Fig1k (top 2 rows appear identical, nearly identical for next 2 rows, another 2 sets of rows appear identical around ~50, and again around ~38, etc), Fig4d (alternating rows on the top have striking similarity during stimulation, rows around 10 are very similar during baseline), Fig5d (many pairs of lines have striking similarities, esp pre and post), Fig6d (strong similarities both on top and bottom).

n's would be helpful in contextualizing this: how many trials from how many neurons from how many animals, and how many neurons were recorded simultaneously. I could only easily find the the total number of neurons in each experiment. But even with more information about n's, there needs to be some other explanation of why so many lines are so similar.

In the heatmaps, we plotted cells ordered by their change in activity during stimulus period (considering 0.5 s bins). We considered change in activity if >20% from baseline. Each row corresponds to an individual cell.

For rats, in the ChR2 group, we recorded 65 cells from 9 animals and in the NpHR group, we recorded 58 cells from 7 animals. For mice, in the ChAT group, we recorded 47 cells from 5 animals; for the VGluT group we recorded 42 cells from 4 animals and for the VGAT group we recorded 53 cells from 6 animals. This is now clearly explained in the methods section. We now provide the reviewer the data tables of the firing rates and and response to optical stimulation.

We also decided to perform additional analysis on the spike data, namely by using the sum of the difference between the spiking activity across units. Given a pair of time series with spiking activity of two units, s_1 , and s_2 , the sum of the difference is given by the sum of the absolute values of the difference (element-wise) between s_1 and s_2 . Considering this, we did not find similarity in our datasets, showing that the recorded cells are very different between them (see Figure 1 below, where we provide histograms of the sum of differences on the spiking activity across units for rats and mice, respectively).

a

b

Figure 1. Histograms of the sum of differences on the spiking activity across units for (a) rats (blue line for ChR2 group and yellow line for NpHR group) and (b) mice (green line for ChAT group, violet line for VGluT group, and red line for VGAT group).

Reviewer #3 (Remarks to the Author):

The authors have addressed my concerns. They may wish to consider including the graphs in their rebuttal in the supplementary data as these results strengthen the paper.

We have now included the additional experiments in Sup. Material (Sup. Fig 11-12) – described in page 16.